# Tri-Comparison Expertise Decision for Drug-Target Interaction Mechanism Prediction

## Abstract

Machine-learned interactions between drugs and human protein targets play a crucial role in efficient and accurate drug discovery. However, the drug-target interaction (DTI) mechanism prediction is actually a multi-class classification problem, which follows a long-tailed class distribution. Existing methods simply address whether interactions can occur and rarely consider the long-tailed DTI mechanism classes. In this paper, we introduce TED-DTI, a novel DTI prediction framework incorporating the divide-and-conquer strategy with tri-comparison options. Specifically, to reduce the learning difficulty of tail classes, we propose an expertise-based divide-and-conquer decision approach that combines the results of multiple independent expertise models for sub-tasks decomposed from the original prediction task. In addition, to enhance the discrimination of similar mechanism classes, we devise a tri-comparison learning strategy that defines the sub-task as the classification of triple options, such as expanding the classification task for classes A and B to include an extra "Neither of them" option. Extensive experiments conducted on various DTI mechanism datasets quantitatively demonstrate the proposed method achieves an approximately 13% performance improvement compared with the other state-of-the-art methods. Moreover, out method exhibits an obvious superiority on the tail classes. Further analysis about the evolvability and generalization of the proposed method reveals the significant potential to be deployed in real-world scenes. Our data and code is included in the Supplementary Materials and will be publicly released after the paper acceptance.

## 1 Introduction

By identifying and developing new pharmaceutical compounds, drug discovery promises to offer breakthrough treatments, improve patient outcomes, and ultimately save lives. In this process, drug-target interactions (DTI) play a critical role, as they provide crucial insights into the mechanisms of action and efficacy of potential drugs, guiding the design and optimization of therapeutic interventions (Keiser et al., 2009; Langedijk et al., 2015). Although the existence of interactions can be reliably confirmed through in vitro binding assays (Liu et al., 2015; 2016; Kang et al., 2016; Yang et al., 2017), the identification process of DTI is significantly time- and resource-consuming (Ullrich et al., 2016) due to the vast search space of chemical compounds. This barrier limits the application of DTI to large-scale disease treatment data. One alternative is using in silico approaches such as docking simulations. Docking simulations consider the 3D structure of drug molecules and targets and identify potential binding sites, which can be experimentally verified. However, the simulation process is still time-consuming (Peska et al., 2017), which typically ranges from a few minutes to several hours. Meanwhile, it cannot be applied if the protein's 3D structure is unknown (Jacob & Vert, 2008; Yamanishi et al., 2008).

In recent years, the rapid advancements in deep learning methods have yielded a significant breakthrough in the computational DTIs, mainly due to the growing availability of extensive biomedical data and domain-specific knowledge. In general, these deep learning-based models (Nath et al., 2018; Lee et al., 2019; Huang et al., 2020a;b; Bai et al., 2023) take the biochemical feature information of drug compounds and target proteins as the input, and output a binary prediction result. These models automatically establish a reasonable and robust mapping relationship between the feature representations and interaction labels, thus enabling large-scale DTI validation within a relatively short time (Gao et al., 2018), thereby accelerating drug discovery processes.

Although deep learning is widely recognized as the most promising method for DTI prediction in current research, existing approaches primarily focus on directly predicting the interactions between drug molecules and target proteins, treating it as a simple binary classification problem. In contrast, the prediction of DTI mechanisms involves multiple mechanism types and exhibits a long-tailed class distribution, which arises with the reason that common action types such as inhibitor (Harding et al., 2018) account for the majority of the available data in the clinical scenes, while rarer interactions such as channel blocker (Harding et al., 2018) are represented in fewer pairs. This uneven distribution leads to some mechanism classes being underrepresented in the datasets, making it challenging for deep models to learn effectively. These current DTI methods overlook and inadequately address this issue, resulting in limited predictive capability for lesser-represented classes. Furthermore, existing long-tailed classification methods (Zhang et al., 2023) leverage the class-balanced re-sampling strategies but often fail to effectively discern the classification boundaries among different mechanism classes as the number of classes increases, thereby limiting their discriminative ability. The decision boundary between any two classes is contaminated with information from other classes, leading to relatively poor overall prediction performance, which undermines the reliability of DTI predictions. Hence, a robust strategy is needed to model multiple clear class boundaries.

To address these challenges, this paper proposes a novel tri-comparison expertise decision method for long-tailed DTI mechanism prediction. First, we adopt the divide-and-conquer strategy and decompose the multi-classification task into pairwise easier-to-learn sub-tasks. Each sub-task is handled by a specific expertise model, thus ensuring that head classes do not dominate the resources of tail classes, thereby rendering the long-tailed task fair and relatively simple to solve. Next, we devise a tri-comparison expertise training strategy for these sub-tasks, which introduces a novel class called $Neither$ and thus expand the classification task from class $\mathcal{A}/\mathcal{B}$ to $\mathcal{A}/\mathcal{B}/Neither$, thereby enhancing the credible decision boundary between classes. Meanwhile, this strategy aids in feature learning for class $\mathcal{A}$ and $\mathcal{B}$ by supplementing a large number of samples from class $Neither$. Finally, a class-balanced decision voting module combines the results from all expertise models, yielding an accurate overall prediction. Experiment results on different datasets show that the proposed method achieves superior performance compared to existing approaches, demonstrating its effectiveness and robustness in handling various real-world scenarios.

The main contributions of this work include (1) introducing a novel Tri-Comparison Expertise Decision approach, namely *TED-DTI*, for long-tailed DTI mechanism prediction; (2) devising a tri-comparison expertise training strategy to enhance the credible decision boundary between classes, along with proposing a class-balanced decision voting module for further expertise combination; (3) conducting extensive experiments to verify the effectiveness and efficiency of TED-DTI, demonstrating its superiority in real-world datasets.

## 2 RELATED WORK

The research aim is to predict long-tailed DTI mechanisms. Hence in this section, we separately elaborate on the related work from DTI prediction and long-tailed classification. Moreover, the classic machine learning strategies, including One-vs-One and One-vs-Rest, are introduced for investigation, although no related work has hitherto been found to apply these strategies to DTI task.

**Drug-Target Interaction.** The latest advancements in artificial intelligence have motivated researchers to employ deep learning methodologies for predicting interactions between drugs and targets. DeepPurpose (Huang et al., 2020a) supports rapid prototyping of customized DTI prediction models with classic encoder-decoder architecture. DeepConv-DTI (Lee et al., 2019) extracts local residue patterns of target protein sequences with a conventional network, similar to the infrastructure of DeepPurpose. MolTrans (Huang et al., 2020b) introduces a knowledge-inspired substructural pattern mining algorithm for enhanced precision and interpretability in DTI prediction. DrugBAN (Bai et al., 2023) utilizes a bilinear attention mechanism to learn pairwise local interactions between drugs and targets and adapt to out-of-distribution data. BINDTI (Peng et al., 2024) leverages a bi-directional intention network to effectively integrate drug and protein features. In addition, BioT5+ (Pei et al., 2024) is a cross-modal pre-trained large language model (LLM) with 252M parameters, designed to enhance cross-modal integration in biology by incorporating chemical knowledge and natural language associations, making it suitable for DTI tasks. We aim to adopt the divide-and-conquer perspective in DTI mechanism prediction task for practical drug discovery.

**Long-tailed Classification.** Long-tailed class imbalance, which is a common problem in practical visual recognition tasks, often limits the practicality of deep network-based recognition models in real-world applications. As a mainstream paradigm in long-tailed learning to address the problem of easily performing poorly on tail classes, class re-balancing (Zhang et al., 2023) seeks to re-balance the negative influence brought by the class imbalance in training sample numbers. This type of methods has three main sub-categories: re-sampling (Ren et al., 2020) aims to re-balance classes by adjusting the number of samples per class in each sample batch for model training; class-sensitive learning (Cao et al., 2019; Cui et al., 2019; Lin et al., 2017; Tan et al., 2020) seeks to particularly adjust the training loss values for various classes to re-balance the uneven training effects caused by the imbalance issue; logit adjustment (Hong et al., 2021; Li et al., 2022) seeks to resolve the class imbalance by adjusting the prediction logits of a class-biased deep model.

**Classic Machine Learning Strategy.** The classic algorithms related to our work include the One-vs-One (OvO) strategy (Allwein et al., 2000; Wu et al., 2003; Galar et al., 2015) and the One-vs-Rest (OvR) strategy (Hong & Cho, 2008). OvO strategy is a common and established technique in machine learning to deal with multi-class classification problems. It consists of dividing the original multi-class problem into easier-to-solve binary sub-tasks considering each possible pair of classes. Similarly, the OvR strategy aims to decompose the original problem, but it does so by splitting the multi-class problem into a binary classification task for each class.

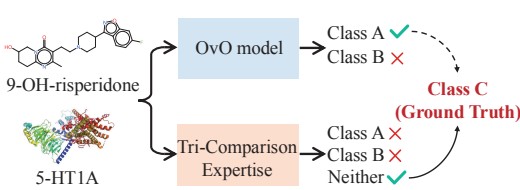

Figure 1: Comparison of the OvO model and the proposed Tri-Comparison Expertise strategy.

However, the OvR strategy exacerbates data imbalance by comparing one class (positive) against all other classes (negative), a challenge that is particularly severe under long-tailed distributions. Moreover, the strict division between positive and negative samples often leads to rigid decision boundaries, resulting in overfitting on head classes and limiting generalization to tail classes. Similarly, while the OvO strategy mitigates data imbalance to some extent by modeling each class pair separately, each classifier is trained only on its corresponding class pair, lacking the ability to handle unrelated samples effectively, making it vulnerable to noise or irrelevant data. Therefore, to address this dilemma, we propose a novel and powerful tri-comparison expertise method to tackle the sub-tasks, with the main differences between the two strategies illustrated in Figure 1. The introduction of the class $Neither$ in the proposed Tri-Comparison Expertise strategy achieves clearer decision boundaries than the original OvO model for classification tasks, while also enriching the dataset with additional samples to obtain more robust feature representations for classes $\mathcal{A}$ and $\mathcal{B}$.

## 3 PROBLEM FORMULATION

In this paper, the task is to determine which mechanism the drug-target pairs obtained from drug compound set and target protein set interact through. For each pair in the dataset, it is assigned a ground truth label $y \in \{1, 2, \ldots, N\}$ where $N$ is the number of DTI mechanism classes[1]. Due to clinical challenges, DTI mechanism prediction is a highly imbalanced multi-class classification task.

For the drug compound $\mathcal{M}$, it is represented by simplified molecular-input line-entry system (SMILES) (Weininger, 1988), which is a 1D sequence describing chemical information of the compound. Due to the lost structural information of 1D sequence, the drug SMILES can also be converted into the corresponding 2D molecular graph. Specifically, a drug molecular graph is defined as $\mathcal{G} = (\mathcal{V}, \mathcal{E})$, where $\mathcal{V}$ is the set of atoms and $\mathcal{E}$ is the set of chemical bonds. For the target protein $\mathcal{T}$, each protein sequence is generally denoted as $\mathcal{T} = \{t_1, t_2, ..., t_o, ..., t_{|\mathcal{T}|}\}$, where each token $t_o$ represents one of the 23 amino acids.

In general, given a drug molecule $\mathcal{M}$ and a protein sequence $\mathcal{T}$, DTI mechanism prediction aims to learn a model to map the joint feature representation space to multi-class mechanism probability vector $p_{(\mathcal{M},\mathcal{T})} \in \mathbb{R}^N$, where $p_{(\mathcal{M},\mathcal{T})}[n] \in [0, 1]$ represents the probability scalar of the $n^{th}$ class.

---

[1]For clarification, important notations in this paper are summarized at Appendix Table 4.

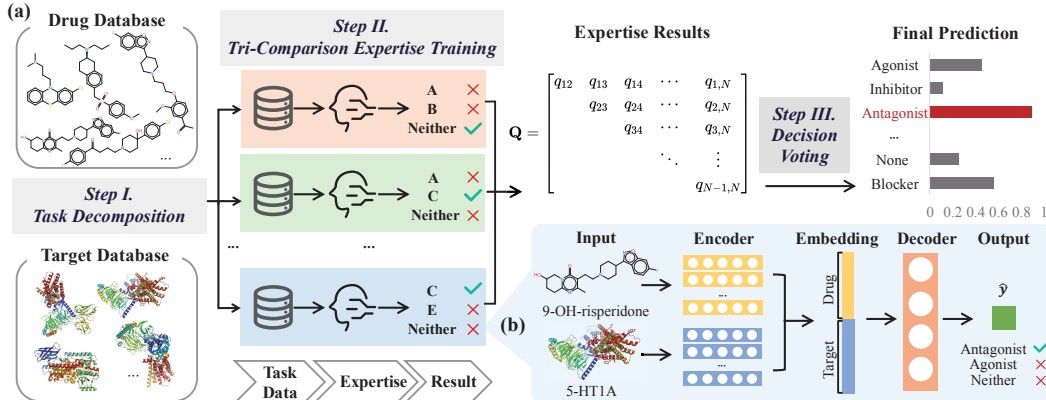

Figure 2: Illustration of the proposed TED-DTI. (a) Pipeline of TED-DTI method. In the training stage, the collected drug and target database are firstly decomposed into different datasets to suit for the corresponding sub-tasks. Then, all the tri-comparison expertise models are trained with the assigned task data and give the prediction results for sub-tasks (A/B/C/E in Step II denote different DTI mechanism classes for simplicity). In the inference stage, all expertise results are combined and thus voted for the original task to determine the probabilities of $N$ mechanisms. (b) A specific example of the expertise model for classifying "Antagonist", "Agonist", or "Neither" of them.

## 4 PROPOSED MODEL

DTI mechanism prediction is a long-tailed multi-class classification problem with similarities among different classes, resulting in fuzzy classification boundaries and difficulty in representation learning of tail classes. In this section, we introduce TED-DTI, a novel tri-comparison expertise decision approach to address the above problems. As shown in Figure 2, TED-DTI is divided into three parts: task decomposition, tri-comparison expertise training and class-balanced decision voting.

### 4.1 TASK DECOMPOSITION

Following the divide-and-conquer strategy, the original task's $N$ classes are decomposed into pairwise sub-tasks before putting in training, and the corresponding datasets are processed simultaneously. Each sub-task aims for the classification of only two classes, such as class $\mathcal{A}$ and $\mathcal{B}$. Ultimately, we obtain $C_N^2 = \frac{N*(N-1)}{2}$ sub-tasks and their respective datasets. The process details can be found at Appendix A.1.

### 4.2 TRI-COMPARISON EXPERTISE TRAINING

To alleviate the the challenges posed by long-tailed distribution for DTI mechanism prediction, a novel class, denoted as $Neither$, is introduced as the third option for each sub-task, alongside the selected classes $\mathcal{A}$ and $\mathcal{B}$. This class contains samples that do not belong to either of the two classes. In the following paper, we will refer to it as $\mathcal{N}_\otimes$ for short.

Specifically, each expertise model is responsible for performing the simple tri-comparison task of determining whether the interaction sample belongs to class $\mathcal{A}$, $\mathcal{B}$ or $\mathcal{N}_\otimes$. The expertise model is based on the classic encoder-decoder architecture. As illustrated in Figure 2b, the encoding module comprises two encoders that process the drug SMILES and target protein sequence, respectively. The decoding module takes the combined drug and protein representations from the encoders as input, and thus predicts its label belonging to $\{\mathcal{A}, \mathcal{B}, \mathcal{N}_\otimes\}$.

**Drug encoder.** Taken the drug SMILES $\mathcal{M}$ as the input, the string is first converted to the molecular graph $\mathcal{G} = (\mathcal{V}, \mathcal{E})$. Atoms in the drug compound are represented as the $f_\mathcal{M}$-dimensional vector $\mathbf{X}_\mathcal{M}^{(0)} \in \mathbb{R}^{|\mathcal{V}| \times f_\mathcal{M}}$ to describe the chemical properties. Vanilla Graph Conventional Network (GCN) (Kipf & Welling, 2016) is adopted as the backbone autoencoder to extract the representation of the

graph $\mathcal{G}$. The initial atom feature $\mathbf{X}_{\mathcal{M}}^{(0)}$ is updated by aggregating the feature vectors of neighborhood atoms through chemical bonds. The propagation mechanism of each GCN layer works as follows:

$$\mathbf{X}_{\mathcal{M}}^{(l+1)} = \sigma(\mathbf{A}\mathbf{X}_{\mathcal{M}}^{(l)}\mathbf{W}_{\mathcal{M}}^{(l)} + \mathbf{b}_{\mathcal{M}}^{(l)}), \tag{1}$$

where $\mathbf{X}_{\mathcal{M}}^{(l)}$, $\mathbf{X}_{\mathcal{M}}^{(l+1)}$ are the hidden atom feature vectors of the $l^{th}$ and $(l+1)^{th}$ GCN layer, respectively; $\mathbf{W}_{\mathcal{M}}^{(l)}$, $\mathbf{b}_{\mathcal{M}}^{(l)}$ are the learnable weight matrix and bias vector of the $l^{th}$ GCN layer; $\mathbf{A}$ represents the adjacency matrix of atoms in the drug graph $\mathcal{G}$; $\sigma(\cdot)$ represents nonlinear activation function, specially ReLU.

After the total number $L_{\mathcal{M}}$ of GCN layers, the weighted sum and max pooling method is applied to the output atom representations $\mathbf{X}_{\mathcal{M}}^{(L_{\mathcal{M}})}$. As a result, the $d_{\mathcal{M}}$-dimensional feature vector $\mathbf{Z}_{\mathcal{M}} \in \mathbb{R}^{d_{\mathcal{M}}}$ of the drug $\mathcal{M}$ is generated for the decoder stage, which is denoted as follows:

$$\mathbf{Z}_{\mathcal{M}} = \text{Pooling}(\mathbf{X}_{\mathcal{M}}^{(L_{\mathcal{M}})}). \tag{2}$$

**Target protein encoder.** Taken the one-dimensional protein sequence $\mathcal{T}$ as the input, the sequence string is first converted to an integer vector as the initialized $f_{\mathcal{T}}$-dimensional embedding $\mathbf{X}_{\mathcal{T}}^{(0)} \in \mathbb{R}^{f_{\mathcal{T}}}$. Then, the 1D CNN model (Kiranyaz et al., 2021) is used to extract the protein representation. The propagation mechanism of each CNN layer works as follows:

$$\mathbf{X}_{\mathcal{T}}^{(l+1)} = \sigma(\text{CNN}(\mathbf{X}_{\mathcal{T}}^{(l)}, d_{in}^{(l)}, d_{out}^{(l)}, k^{(l)})), \tag{3}$$

where $\mathbf{X}_{\mathcal{T}}^{(l)}$, $\mathbf{X}_{\mathcal{T}}^{(l+1)}$ are the hidden feature vectors of the $l^{th}$ and $(l+1)^{th}$ CNN layer, respectively; $d_{in}^{(l)}$, $d_{out}^{(l)}$, $k^{(l)}$ are the number of channels in the input, number of channels produced by the convolution and the convolving kernel size of the $l^{th}$ CNN layer; $\sigma(\cdot)$ represents nonlinear activation function, specially ReLU.

After the total number $L_{\mathcal{T}}$ of CNN layers, the $d_{\mathcal{T}}$-dimensional feature vector of target protein $\mathbf{Z}_{\mathcal{T}} \in \mathbb{R}^{d_{\mathcal{T}}}$, which is equal to $\mathbf{X}_{\mathcal{T}}^{(L_{\mathcal{T}})}$, is generated for the decoder stage.

**Decoder for DTI prediction.** As the decoder, a total of $L$-layer Multi-Layer Perceptron (MLP) (Murtagh, 1991) network uses the joint representation $\mathbf{Z}^{(0)} \in \mathbb{R}^{d_{\mathcal{M}}+d_{\mathcal{T}}}$ generated by the combination of $\mathbf{Z}_{\mathcal{M}}$ and $\mathbf{Z}_{\mathcal{T}}$ to predict the probabilities of the final three classes $p \in \mathbb{R}^3$, which is calculated as follows:

$$\mathbf{Z}^{(l+1)} = \sigma(\text{MLP}(\mathbf{Z}^{(l)}, \mathbf{W}^{(l)}, \mathbf{b}^{(l)})), \hat{p} = \text{Softmax}(\mathbf{Z}^{(L)}), \tag{4}$$

where $\mathbf{Z}^{(l)}$, $\mathbf{Z}^{(l+1)}$ are the hidden feature vectors of the $l^{th}$ and $(l+1)^{th}$ MLP layer, respectively; $\mathbf{W}^{(l)}$, $\mathbf{b}^{(l)}$ are the learnable weight matrix and bias vector of the $l^{th}$ MLP layer; $\sigma(\cdot)$ represents nonlinear activation function, specially ReLU; $\text{Softmax}(\cdot)$ represents nonlinear activation function; $\hat{p}$ represents the probability vector of three prediction classes, i.e. $\mathcal{A}/\mathcal{B}/\mathcal{N}_{\otimes}$.

**Training loss.** After that, the training loss for each tri-comparison expertise model is calculated as follows: $\mathcal{L} = -\frac{1}{3}\sum_{n=1}^{3} p_n \log(\hat{p}_n)$, where $\hat{p}_n$, $p_n$ denotes the probability and true label of the $n^{th}$ class, respectively. Note that, the goal of each model is to classify the three classes $\mathcal{A}/\mathcal{B}/\mathcal{N}_{\otimes}$.

## 4.3 Class-balanced Decision Voting

During the inference stage, the triple-option prediction results obtained from all expertise models cannot be simply combined with the voting strategy as introduced in traditional OvO. To this end, we propose a novel class-balanced decision voting strategy to effectively amalgamate the predictions of all these expertise models. Specifically, we obtain $C_N^2 = \frac{N*(N-1)}{2}$ initial prediction results $\mathbf{Q} \in \mathbb{R}^{\frac{N*(N-1)}{2}}$ from the expertise models, which is defined as follows:

$$\mathbf{Q} = (q_{12}, q_{13}, \cdots, q_{1N}, q_{23}, \cdots, q_{2N}, \cdots, q_{N-1,N}), \tag{5}$$

where $q_{i,j} \in \{-1, 0, 1\}$ represents the prediction result for the sub-task of classifying class $i$ and class $j$.

Next, the final voting vector $\mathbf{Y} \in \mathbb{R}^N$ of $N$ classes is updated with three possible outputs based on the reward-penalty strategy as follows:

- if $q_{i,j}$ is 0, which indicates that class $i$ is the output label, the reward $\beta_R$ is allocated to the voting score of class $i$, denoted as $\mathbf{Y}_i$;

- if $q_{i,j}$ is 1, which indicates that class $j$ is the output label, the reward $\beta_R$ is allocated to the voting score of class $j$, denoted as $\mathbf{Y}_j$;

- if $q_{i,j}$ is -1, which indicates that class $\mathcal{N}_\otimes$ is the output label, the penalty score is allocated to both $\mathbf{Y}_i$ and $\mathbf{Y}_j$. To compute the penalty score, a class-balanced weight vector $\mathbf{H} \in \mathbb{R}^N$ is multiplied with the base penalty score $\beta_P$. Specifically, $\mathbf{H}$ assigns a weight to each class to ensure a fair evaluation of their contributions. The weight for class $n$, denoted as $\mathbf{H}_n$, is determined by the formula $\mathbf{H}_n = \frac{\frac{1}{S_n}}{\sum_{k=1}^{N} \frac{1}{S_k}}$, where $S_n$ represents the sample number of class $n$, and $N$ is the total number of classes.

After iterating through the predictions of all expertise models, the vote scores for all classes are tallied and the class with the highest score is selected as the final prediction $\hat{y}$. Detailed voting algorithm can be found at Appendix Algorithm 2.

### 4.4 INFERENCE PROCESS OF TED-DTI

Given the above expertise training and class-balanced decision voting modules, the inference process of TED-DTI is thus illustrated for a clear understanding as follows:

---

**Algorithm 1** Example for the inference process of TED-DTI.

---

**Input:** Drug $\mathcal{M}$ with its molecular graph $\mathcal{G} = (\mathcal{V}, \mathcal{E})$; Target protein $\mathcal{T}$ with its sequence $\{t_1, t_2, ..., t_o, ..., t_{|\mathcal{T}|}\}$; The parameter set of all the trained expertise models $\boldsymbol{\theta}$.
**Output:** Final prediction DTI mechanism class $\hat{y}_{(\mathcal{M}, \mathcal{T})}$.

1: Initialize the drug feature $\mathbf{X}_{\mathcal{M}}^{(0)}$ and protein feature $\mathbf{X}_{\mathcal{T}}^{(0)}$ with the corresponding bio-knowledge.
2: **for** each sub-task for classifying class pair $(i, j)$ **do**
3: $\quad \theta_G, \theta_C, \theta_M \leftarrow \boldsymbol{\theta}_{i,j}$
4: $\quad \mathbf{Z}_{\mathcal{M}} \leftarrow \text{GCN}(\mathbf{X}_{\mathcal{M}}^{(0)}, \mathcal{G}, \theta_G);$
5: $\quad \mathbf{Z}_{\mathcal{T}} \leftarrow \text{CNN}(\mathbf{X}_{\mathcal{T}}^{(0)}, \theta_C);$
6: $\quad q_{i,j} \leftarrow \text{MLP}(\mathbf{Z}_{\mathcal{M}}, \mathbf{Z}_{\mathcal{T}}, \theta_M).$
7: **end for**
8: Class-balanced decision voting for all expertise results $\mathbf{Q}$ to get the voting results $\mathbf{Y}$ for all $N$ classes.
9: **return** $\hat{y}_{(\mathcal{M}, \mathcal{T})} \leftarrow \text{argmax}(\mathbf{Y})$

---

### 4.5 THEORETICAL ANALYSIS

The tri-comparison strategy provides a comprehensive solution for multi-class classification, particularly in long-tailed tasks like DTI mechanism prediction. By integrating decision boundary theory and error decomposition, it enhances both performance and generalization.

In traditional binary classification, decision boundaries (e.g., $f_{i,j}(x)$ for classes $i$ and $j$) often suffer from noise and bias due to overlapping regions from unrelated samples, especially in long-tailed distributions. The tri-comparison strategy addresses this by introducing class $\mathcal{N}_\otimes$, with a new decision boundary $f_{\mathcal{N}_\otimes}(x)$, creating three distinct regions: $\mathbb{R}^d = \{x : f_i(x) > f_{\mathcal{N}_\otimes}(x)\} \cup \{x : f_j(x) > f_{\mathcal{N}_\otimes}(x)\} \cup \{x : f_{\mathcal{N}_\otimes}(x) > \max(f_i(x), f_j(x))\}$ This refinement in decision boundaries reduces the noise caused by ambiguous samples, ensuring clearer separation between classes and laying a foundation for improved classification accuracy.

Furthermore, in binary classification, the overall error $\epsilon_{\text{binary}}$ is dominated by the false negative rate of minority classes and the false positive rate of majority classes. By explicitly isolating unrelated samples into class $\mathcal{N}_\otimes$, the classification error is redefined as $\epsilon_{\text{tri}} = \epsilon_{\text{false positive}} + \epsilon_{\text{false negative}} + \epsilon_{\mathcal{N}_\otimes}$. This separation reduces the overlap between positive and negative classes, significantly lowering $\epsilon_{\text{false positive}}$ and $\epsilon_{\text{false negative}}$, and consequently decreasing the total error. The tri-comparison strategy thus moves beyond simple noise reduction, actively addressing imbalances in class representation to improve classification reliability.

## 5 EXPERIMENTS

### 5.1 EXPERIMENTAL SETTINGS

**Datasets**. The International Union of Basic and Clinical Pharmacology/British Pharmacological Society Guide to PHARMACOLOGY database (GtoPdb) (Harding et al., 2018) is used for the DTI mechanism prediction experiments. Due to the presence of numerous missing essential items in the original dataset, we first preprocess the dataset before putting it into training. After that, we get 13,381 data pairs in the (drug SMILES, target sequence, DTI mechanism class) triplet format, in which the former two are used as the model input and the latter as the ground truth label. Figure 3 shows the eight DTI mechanism classes of GtoPdb dataset and the corresponding sample numbers. The details of data preprocessing are provided at Appendix B.1.

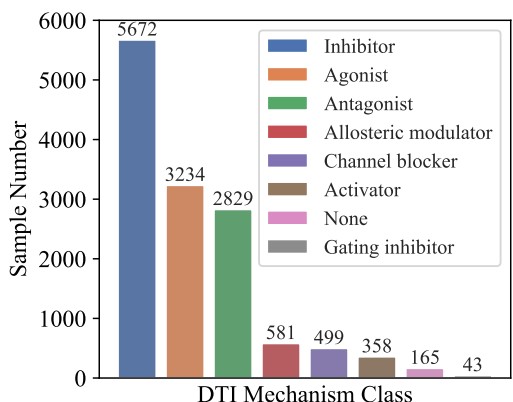

Figure 3: Detailed information of GtoPdb dataset and its corresponding DTI mechanism classes.

Moreover, as a large and open-access drug discovery database, ChEMBL (Mendez et al., 2018) contains abundant DTI information from the real-world scenes. However, the samples with complete mechanism label information are limited. After screening, 829 data triplets from ChEMBL are obtained as the real-world data for independent and challenging external test and thus preprocessed with the same strategy as GtoPdb.

For further validation of generalizability, the samples related to G-protein coupled receptors (GPCRs) is separately collected from GtoPdb, namely GtoPdb-GPCRs. Essentially, GtoPdb-GPCRs is a subset of GtoPdb. As the primary target receptor of human body (about 50% of drugs currently on the market), GPCRs (Overington et al., 2006) have been studied for the agonistic, antagonistic or inactive mechanisms against different drugs. Ultimately, 5,111 GPCRs triplets are obtained for generalizability validation. Detailed information for dataset details is provided at Appendix B.2.

**Metrics**. To evaluate the method performance on DTI mechanism prediction, Accuracy and F1 score are employed for model ability to provide a comprehensive assessment for the multi-class classification task, with Accuracy measuring overall correctness and F1 score balancing precision and recall to address potential class imbalances. Furthermore, for few-sample problem, we validate only the classification performance between extreme tail class and all the other classes, thus framing the task as a binary prediction and utilizing AUROC for robust predictions of tail classes.

**Implementation Details**. To accurately evaluate model performance and prevent overfitting, we use 5-fold cross-validation to train models only used the GtoPdb training set, and we evaluate model performance on both the GtoPdb test set (internal test) and the entire ChEMBL dataset (external test) using the trained models. Adam optimizer is adopted to optimize all parameters of the model with a learning rate of 0.001. The batch size is setting to 32. The Cross Entropy loss function is used to measure model performance in the expertise training stage. Details are provided at Appendix B.4.

**Baselines**. To verify the effectiveness of TED-DTI, we compare it with the SOTA methods from three perceptives: **Drug-Target Interaction**. Five current advanced deep learning methods are adopted, including DeepPurpose, DeepConv-DTI, MolTrans, DrugBAN, BINDTI, and a cross-modal LLM BioT5+. Note that we implement the pair combination of 7 drug encoders and 7 target encoders to display the performance of DeepPurpose. **Long-tailed Learning**. Long-tailed methods are selected base on accessible source codes and no non-trivial modifications. Then, eight methods are empirically evaluated in this paper, including Balanced Softmax, Weighted Softmax, Focal Loss, Equalization loss (ESQL), LADE, Class-balanced loss (CB), GCL, LDAM. **Classic Machine**

Table 1: Performance comparison on the GtoPdb and ChEBML datasets. "DTI" indicates the drug-target interaction methods; "LTL" indicates long-tailed learning based methods; "CML" indicates classic machine learning methods (OvO & OvR). All results are presented as "mean$_{\pm\text{standard deviation}}$" and the best result for each dataset and metric is marked in **bold**. $\Delta$ in the last line indicates the performance improvement (in %) of our method compared to the suboptimal method.

| Type | Methods | Reference | GtoPdb | | ChEBML | |
|---|---|---|---|---|---|---|
| | | | Accuracy↑ | F1 score↑ | Accuracy↑ | F1 score↑ |
| DTI | DeepConv-DTI | (Lee et al., 2019) | $0.898_{\pm0.022}$ | $0.791_{\pm0.032}$ | $0.922_{\pm0.028}$ | $0.634_{\pm0.098}$ |
| | DeepPurpose | (Huang et al., 2020a) | $0.907_{\pm0.008}$ | $0.804_{\pm0.031}$ | $0.939_{\pm0.006}$ | $0.559_{\pm0.041}$ |
| | MolTrans | (Huang et al., 2020b) | $0.901_{\pm0.004}$ | $0.792_{\pm0.018}$ | $0.873_{\pm0.011}$ | $0.577_{\pm0.048}$ |
| | DrugBAN | (Bai et al., 2023) | $0.908_{\pm0.004}$ | $0.803_{\pm0.016}$ | $0.959_{\pm0.005}$ | $0.691_{\pm0.076}$ |
| | BINDTI | (Peng et al., 2024) | $0.908_{\pm0.002}$ | $0.806_{\pm0.028}$ | $0.934_{\pm0.006}$ | $0.676_{\pm0.029}$ |
| LTL | Weighted Softmax | - | $0.911_{\pm0.003}$ | $0.813_{\pm0.014}$ | $0.947_{\pm0.003}$ | $0.591_{\pm0.070}$ |
| | Focal Loss | (Lin et al., 2017) | $0.914_{\pm0.003}$ | $0.808_{\pm0.023}$ | $0.944_{\pm0.008}$ | $0.610_{\pm0.076}$ |
| | CB | (Cui et al., 2019) | $0.913_{\pm0.004}$ | $0.809_{\pm0.011}$ | $0.946_{\pm0.006}$ | $0.651_{\pm0.058}$ |
| | LDAM | (Cao et al., 2019) | $0.910_{\pm0.005}$ | $0.813_{\pm0.018}$ | $0.945_{\pm0.005}$ | $0.618_{\pm0.057}$ |
| | ESQL | (Tan et al., 2020) | $0.911_{\pm0.004}$ | $0.808_{\pm0.021}$ | $0.947_{\pm0.005}$ | $0.563_{\pm0.083}$ |
| | Balanced Softmax | (Ren et al., 2020) | $0.906_{\pm0.007}$ | $0.794_{\pm0.022}$ | $0.935_{\pm0.014}$ | $0.535_{\pm0.045}$ |
| | LADE | (Hong et al., 2021) | $0.915_{\pm0.004}$ | $0.804_{\pm0.021}$ | $0.952_{\pm0.005}$ | $0.699_{\pm0.097}$ |
| | GCL | (Li et al., 2022) | $0.913_{\pm0.004}$ | $0.816_{\pm0.016}$ | $0.945_{\pm0.010}$ | $0.605_{\pm0.044}$ |
| CML | SVM-based OvO | (Cortes & Vapnik, 1995) | $0.831_{\pm0.036}$ | $0.682_{\pm0.039}$ | $0.856_{\pm0.038}$ | $0.507_{\pm0.049}$ |
| | GCN-based OvO | (Kipf & Welling, 2016) | $0.916_{\pm0.004}$ | $0.812_{\pm0.030}$ | $0.955_{\pm0.007}$ | $0.648_{\pm0.129}$ |
| | GCN-based OvR | (Kipf & Welling, 2016) | $0.887_{\pm0.010}$ | $0.732_{\pm0.049}$ | $0.910_{\pm0.015}$ | $0.566_{\pm0.051}$ |
| Ours | TED-DTI | - | $\mathbf{0.924_{\pm0.004}}$ | $\mathbf{0.834_{\pm0.012}}$ | $\mathbf{0.961_{\pm0.003}}$ | $\mathbf{0.789_{\pm0.040}}$ |
| | $\Delta$ | - | +0.87% | +2.21% | +0.21% | +12.88% |

**Learning strategy**. The OvO methods are implemented using different backbone models, including Support Vector Machine (SVM) and GCN[2]. Similarly, the OvR method is implemented with GCN.

## 5.2 QUANTITATIVE ANALYSIS

**Performance Comparison with SOTAs.** As illustrated in Table 1, the performance results for DTI mechanism prediction on the GtoPdb dataset indicate that TED-DTI outperforms all comparative methods across DTI, LTL, and OvO perspectives in terms of Accuracy and F1 score, demonstrating its effectiveness in DTI mechanism prediction. Furthermore, to demonstrate the robustness of the TED-DTI method on real-world and out-of-domain data, 829 data triplets from the ChEMBL dataset are used as an independent test set to evaluate the model trained on the GtoPdb dataset. The results, as shown in Table 1, indicate that TED-DTI still outperforms other comparative methods, thus confirming that the proposed method is highly generalizable in real scenarios. Remarkably, TED-DTI achieves a notably high F1 score on the ChEMBL dataset, with a substantial improvement of approximately 13% (from 0.699 to 0.789) compared to other methods, indicating that other models struggle with the out-of-domain data from the ChEMBL dataset. In contrast, TED-DTI employs a tri-comparison expertise strategy that effectively mitigates the impact of cross-domain data on model generalization, leading to a considerable performance improvement on the ChEMBL dataset. Notably, Table 2 shows that TED-DTI, with only 1/25 of the parameters, still outperforms BioT5+. This demonstrates that even with a significantly smaller model size, our approach achieves superior performance, highlighting its balance between parameter efficiency and task accuracy.

**Improvements on Few-sample Class.** Further, we focus on the few-sample problem in DTI mechanism prediction. Specifically, the data of certain DTI mechanism (such as Gating Inhibitor) is highly scarce, which hinders the application of deep learning methods in real scenes. In the validation experiment of few-sample class, the test data of the "Gating Inhibitor" class (only 0.3% of the whole dataset) is used as the extreme tail class for the binary classification task with all other classes. Figure 4a shows the performance of TED-DTI and other baseline methods. TED-DTI surpasses other baselines by achieving the highest average AUROC score of 0.914. We also have the following observations: (1) Compared with DTI methods which only consider whether the interaction will occur,

---

[2]This implementation shares the same network architecture as our method, except that the output of each sub-model does not include class $\mathcal{N}_{\otimes}$.

Table 2: Performance and parameter comparison with cross-modal LLM BioT5+. Note that the number of trained parameters for TED-DTI is presented as the total sum of all 28 sub-task models.

| Methods | Reference | #Parameters | GtoPdb | | ChEBML | |
|---|---|---|---|---|---|---|
| | | | Accuracy | F1 score | Accuracy | F1 score |
| BioT5+ | (Pei et al., 2024) | 252M | $0.920_{\pm0.003}$ | $0.829_{\pm0.022}$ | $0.954_{\pm0.002}$ | $0.767_{\pm0.018}$ |
| TED-DTI | - | 10M | $\mathbf{0.924_{\pm0.004}}$ | $\mathbf{0.834_{\pm0.012}}$ | $\mathbf{0.961_{\pm0.003}}$ | $\mathbf{0.789_{\pm0.040}}$ |

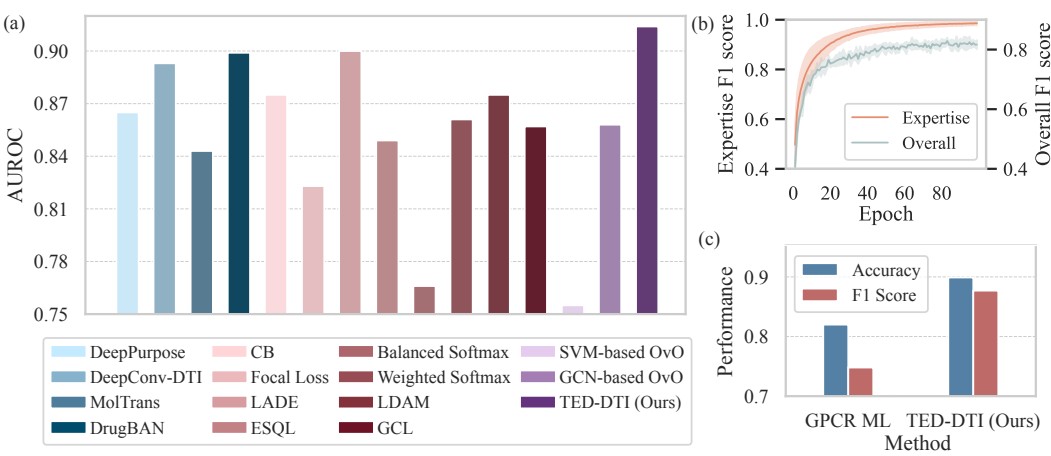

Figure 4: Illustration of the ability to address tail class, evolvability, and generalization of TED-DTI. (a) Performance comparison of few-sample class "Gating inhibitor" (account for 0.3%) on the test set of GtoPdb dataset. (b) Performance trends of expertise models and the overall prediction. (c) Generalization validity of TED-DTI for GPCRs DTI task on the GtoPdb-GPCRs dataset.

TED-DTI achieves better performances than all these baselines, which indicates that the discrimination of head classes and tail classes needs to be considered and treating each class equally can not extract adequate information from known classes; (2) Compared with LTL methods which focus on balancing all classes uniformly, TED-DTI has a varying degree of improvement than all these methods, which demonstrates that the complexity of the task can be reduced through task decomposition and thus there is a significant enhancement on the feature learning of the tail class; (3) Compared with OvO methods which also adopt the divide-and-conquer strategy, TED-DTI significantly exceeds all the OvO baselines, which implies that the devise of class $\mathcal{N}_{\otimes}$ can effectively determine the decision boundaries of mechanism classes and thus improve the prediction performance.

**Continuous Evolvability Analysis.** To validate that TED-DTI method has the capacity for continuous evolution, we present the test performance of each expertise model during the initial 100 epochs, along with the overall prediction performance achieved through decision voting on the test set. As shown in Figure 4b, the changing trend of the overall prediction performance varies with the training epoch of the single expertise model. As the number of training epochs increases, the expertise models gradually converge, resulting in consistent improvement in overall prediction performance. On the other hand, when the performance of the expertise models reaches a bottleneck, the growth in overall prediction performance also slows down, indicating that this overall performance is constrained by the predictive capabilities of the expertise models.

**Generalization on Similar Tasks.** To validate the generalization capabilities of TED-DTI on other class-imbalanced DTI tasks, we apply this strategy to the GPCRs DTI (Overington et al., 2006) problem. This task, while similar, deals with a different scale and investigates agonistic, antagonistic, or inactive mechanisms in response to various drugs. Figure 4c shows the performance comparison of TED-DTI method and GPCR ML (Oh et al., 2022) on the GtoPdb-GPCRs dataset. Notably, TED-DTI demonstrates substantial improvements in multi-classification metrics, with accuracy rising from 0.820 to 0.889 and the F1 score increasing from 0.748 to 0.877, thereby emphasizing the remarkable potential for application across various tasks and domains in the real-world scenes.

## 5.3 Ablation Study

To investigate the necessity of each component in TED-DTI, we conduct several comparisons between TED-DTI and its variants on the test set: **TED-DTI without class $\mathcal{N}_{\otimes}$ (w/o $\mathcal{N}_{\otimes}$)** excludes class $Neither$, and directly adopts the classification of class $\mathcal{A}$ and class $\mathcal{B}$ as the training objective of expertise model. **TED-DTI without class-balanced penalty (w/o CP)** eliminates the class-balanced penalty step applied to the voting results, and thus resets to the vanilla vote mechanism.

As illustrated in Table 3, when these basic components of TED-DTI have been removed, the performances of the corresponding variants on the test dataset exhibit a significant drop, indicating that these components all contribute to the performance.

Table 3: Ablation results on the crucial components of TED-DTI.

| Methods | GtoPdb | | ChEBML | |
|---|---|---|---|---|
| | Accuracy | F1 score | Accuracy | F1 score |
| w/o $\mathcal{N}_{\otimes}$ | $0.916_{\pm 0.004}$ | $0.812_{\pm 0.030}$ | $0.955_{\pm 0.007}$ | $0.648_{\pm 0.129}$ |
| w/o CP | $0.920_{\pm 0.005}$ | $0.829_{\pm 0.013}$ | $0.957_{\pm 0.006}$ | $0.768_{\pm 0.117}$ |
| TED-DTI | $\mathbf{0.924_{\pm 0.004}}$ | $\mathbf{0.834_{\pm 0.012}}$ | $\mathbf{0.961_{\pm 0.003}}$ | $\mathbf{0.789_{\pm 0.040}}$ |

When the classification for class $\mathcal{N}_{\otimes}$ has been removed from the sub-task, the performance of the corresponding variant significantly declines. Especially, the observation that the F1 score declines from 0.789 to 0.648 on the ChEBML dataset indicates the prediction performance is boosted mostly by the class $\mathcal{N}_{\otimes}$ and thus the design of class $\mathcal{N}_{\otimes}$ brings the improvement of the discrimination between mechanism classes and more expressive representations. Moreover, after the removal of the class-balanced penalty from the voting module, these performance metrics exhibit varying degrees of decline, particularly with a noticeable decrease in F1 score on the ChEBML dataset, which indicates that: (1) the design of the class-balanced penalty makes the overall voting stage more favorable for tail classes; (2) despite removing the penalty but retaining the class $\mathcal{N}_{\otimes}$, there is no substantial performance drop. This indicates that the class balance penalty weight is not essential to address the long-tail problem, but indeed helps to balance the importance of different classes and thus brings improvements. Furthermore, TED-DTI w/o CP (i.e., with class weights all set to 1) still outperforms the other baselines, reinforcing that class $\mathcal{N}_{\otimes}$ is the core component of the proposed method.

## 6 Limitation and Future Work

The divide-and-conquer strategy requires decomposing the original task into sub-tasks. As the number of classes $N$ increases, the number of expertise models that need to be trained grows exponentially, potentially leading to a critical resource overload. Furthermore, even though complex tasks are broken down into relatively simpler sub-tasks, issues such as class imbalance during the training process of the expertise models can still arise. These challenges may create performance bottlenecks, ultimately hindering further optimization of overall performance.

Future work will optimize TED-DTI with efficient learning algorithms to reduce resource use and enable dynamic selection of expertise models in constrained environments. We will investigate data augmentation techniques to address data imbalance and ensure balanced performance across classes. Finally, we will explore applications in other domains to better serve real-world scenarios, demonstrating the broader impact and versatility of our approach.

## 7 Conclusion

In this paper, we present TED-DTI, a tri-comparison expertise decision method designed specifically for long-tailed DTI mechanism prediction. TED-DTI employs a divide-and-conquer strategy, utilizing outputs of various independent expertise models to tackle sub-tasks decomposed from the original long-tailed problem. Moreover, we introduce a novel class, denoted as $Neither$, specifically designed to facilitate the tri-comparison sub-task. Additionally, a class-balanced decision module is designed to seamlessly integrate the results from all expertise models. Extensive experimental results reveal that TED-DTI outperforms other baseline methods, demonstrating that the incorporation of the class $Neither$ significantly enhances the discrimination among similar mechanism classes and yields more effective and robust feature representations for tail classes. Furthermore, a thorough exploration of the evolvability and generalization capabilities of TED-DTI underscores its practical utility and effectiveness for deployment in real-world scenarios.

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

# A DETAIL OF TED-DTI METHOD

Detailed information about the proposed TED-DTI method is provided as follows. First, as shown in Table 4, the key notations and the corresponding definitions are summarized for clarification. Then, the first step of TED-DTI about task decomposition is elaborated. Finally, the algorithm of class-balanced decision voting module is provided for supplement the introduction of the main paper.

Table 4: Notations and Definitions.

| Notation | Definition |
|---|---|
| $N$ | The number of DTI mechanism classes. |
| $\mathcal{A}, \mathcal{B}$ | The abbreviation for the simplification of the mechanism class pair for the sub-task. |
| $\mathcal{N}_\otimes$ | The introduced third option for each sub-task, alongside the two selected classes. |
| $\mathcal{G}$ | $\mathcal{G} = (\mathcal{V}, \mathcal{E})$ indicates a molecular graph, and $\mathcal{V}, \mathcal{E}$ indicate the set of atoms and the set of chemical bonds, respectively. |
| $\mathcal{V}$ | The set of atoms of $\mathcal{G}$. |
| $\mathcal{E}$ | The set of chemical bonds of $\mathcal{G}$. |
| $f_\mathcal{M}$ | The input feature dimensional of atoms in the drug $\mathcal{M}$. |
| $f_\mathcal{T}$ | The input feature dimensional of the protein $\mathcal{T}$. |
| $d_\mathcal{M}$ | The output feature dimensional of the molecular graph $\mathcal{G}$ of drug $\mathcal{M}$. |
| $d_\mathcal{T}$ | The output feature dimensional of the protein $\mathcal{T}$. |
| $\mathbf{X}_\mathcal{M}^{(0)}$ | The initial atom feature for molecule $\mathcal{M}$. $\mathbf{X}_\mathcal{M}^{(0)} \in \mathbb{R}^{|\mathcal{V}| \times f_\mathcal{M}}$. |
| $\mathbf{X}_\mathcal{M}^{(l)}$ | The input atom feature for molecule $\mathcal{M}$ of GCN layer $l$. |
| $\mathbf{Z}_\mathcal{M}$ | The output graph feature for molecule $\mathcal{M}$. $\mathbf{Z}_\mathcal{M} \in \mathbb{R}^{d_\mathcal{M}}$. |
| $\mathbf{X}_\mathcal{T}^{(0)}$ | The initial feature for target protein $\mathcal{T}$. $\mathbf{X}_\mathcal{T}^{(0)} \in \mathbb{R}^{f_\mathcal{T}}$. |
| $\mathbf{X}_\mathcal{T}^{(l)}$ | The input feature for target protein $\mathcal{T}$ of 1D CNN layer $l$. |
| $\mathbf{Z}_\mathcal{T}$ | The output feature for target protein $\mathcal{T}$. $\mathbf{Z}_\mathcal{T} \in \mathbb{R}^{d_\mathcal{T}}$. |
| $\mathbf{Z}^{(0)}$ | The joint representation generated by the combination of $\mathbf{Z}_\mathcal{M}$ and $\mathbf{Z}_\mathcal{T}$ and meanwhile the input feature of the predictor module. $\mathbf{Z}^{(0)} \in \mathbb{R}^{d_\mathcal{M}+d_\mathcal{T}}$. |
| $\mathbf{Z}^{(l)}$ | The input feature of MLP layer $l$. |
| $\mathbf{Q}$ | The initial prediction results obtained from the expertise models. $\mathbf{Q} \in \{-1, 0, 1\}^{\frac{N*(N-1)}{2}}$. |
| $q_{i,j}$ | The prediction result for the sub-task of classifying class $i$ and class $j$. $q_{i,j} \in \{-1, 0, 1\}$. |
| $\mathbf{Y}$ | The final voting results of $N$ classes. $\mathbf{Y} \in \mathbb{R}^N$. |
| $\mathbf{H}$ | The class-balanced weight vector for penalty score of different classes. $\mathbf{H} \in \mathbb{R}^N$. |
| $\beta_R$ | The base reward score for expertise predictions. |
| $\beta_P$ | The base penalty score for expertise predictions. |

## A.1 TASK DECOMPOSITION

The original DTI mechanism prediction task is denoted as a multi-classification task with $N$ classes. Each sub-task aims for the classification of only two classes, resulting in a total of $C_N^2 = \frac{N*(N-1)}{2}$ sub-tasks. Each sub-task is trained by a dedicated expertise model to extract knowledge related to the corresponding two classes. To effectively determine the classification boundaries of mechanism classes, we introduce an additional class $\mathcal{N}_\otimes$ for samples that do not belong to the selected two classes.

Taking the classification of the two interaction mechanisms $\mathcal{A}$ and $\mathcal{B}$ as an example, we first extract the class-related samples from the original dataset $\mathcal{D}$, and denote them as $\mathcal{D}_\mathcal{A}, \mathcal{D}_\mathcal{B}$. Meanwhile, the samples $\mathcal{D}_{\mathcal{N}_\otimes}$ belonging to class $\mathcal{N}_\otimes$ are randomly sampled from the dataset excluding class $\mathcal{A}$ and $\mathcal{B}$, with the total number of samples equal to the mean number of samples in class $\mathcal{A}$ and $\mathcal{B}$, i.e., $\frac{|\mathcal{D}_\mathcal{A}|+|\mathcal{D}_\mathcal{B}|}{2}$. This sampling strategy is adopted to achieve a balanced three-class prediction task and mitigate the severe long-tail problem that may exist in the original dataset. Ultimately, the training

dataset for this task $\mathcal{D}_{A/B}$ is the combination of three datasets $\mathcal{D}_A$, $\mathcal{D}_B$, $\mathcal{D}_{\mathcal{N}_\otimes}$. The relationship of the three datasets and the ground truth label $y_{A/B}$ are represented as follows:

$$\mathcal{D}_{\mathcal{N}_\otimes} \subseteq \mathcal{D} - (\mathcal{D}_\mathcal{A} \cup \mathcal{D}_\mathcal{B}), \mathcal{D}_{\mathcal{A}/\mathcal{B}} = \{\mathcal{D}_\mathcal{A}, \mathcal{D}_\mathcal{B}, \mathcal{D}_{\mathcal{N}_\otimes}\}, y_{\mathcal{A}/\mathcal{B}} \in \{\mathcal{A}, \mathcal{B}, \mathcal{N}_\otimes\}. \tag{6}$$

### A.2 Algorithm of Class-balanced Decision Voting

Here, the detailed algorithm of the class-balanced decision voting module is shown as follows:

---
**Algorithm 2** Class-balanced decision voting process for the final prediction.

---
**Input:** The initial prediction results of all the expertise models $\mathbf{Q}$; $N$ is the number of the DTI mechanism classes; $\beta_R$ is the base reward score; $\beta_P$ is the base penalty score; $\mathbf{H}$ is the balanced penalty weight vector.
**Output:** Final prediction label $\hat{y}$.
1: $\mathbf{Y} \leftarrow (0, ..., 0)_N$; //Initialized vote results
2: **for** $q_{i,j}$ in $\mathbf{Q}$ **do**
3:     **if** $q_{i,j}$ is 0 **then**
4:         $\mathbf{Y}_i \leftarrow \mathbf{Y}_i + \beta_R$;
5:     **else if** $q_{i,j}$ is 1 **then**
6:         $\mathbf{Y}_j \leftarrow \mathbf{Y}_j + \beta_R$;
7:     **else if** $q_{i,j}$ is -1 **then**
8:         $\mathbf{Y}_i \leftarrow \mathbf{Y}_i - \beta_P \cdot \mathbf{H}_i$;
9:         $\mathbf{Y}_j \leftarrow \mathbf{Y}_j - \beta_P \cdot \mathbf{H}_j$;
10:     **end if**
11: **end for**
12: **return** $\hat{y} \leftarrow \mathrm{argmax}(\mathbf{Y})$

---

## B Detail of Experiment

For further analysis, the details about datasets, implementations, and experimental results are provided. First, the dataset preprocessing steps and thus the statistical details of the preprocessed three datasets are outlined. Then, comprehensive performance comparisons of the DeepPurpose baseline method are presented. Finally, the implementation details of the proposed method and the compute resources of all the experiments are elaborated.

### B.1 Dataset Preprocessing

For experiments of DTI mechanism prediction, the following steps are applied to the two datasets GtoPdb and ChEBML before put into training or test:

**Domain Filtering.** only retain drug-target pairs that are relevant to humans and have complete field information, that is, drug SMILES and target protein identifier SwissProt;

**Validity Check.** use RDKit package (Landrum, 2016) to determine whether the drug SMILES is illegal;

**Data Match.** match the corresponding protein sequence in the UniProt database (Consortium, 2022) according to the SwissProt identifier;

**Statistical Analysis.** analyze the DTI mechanism classes field (prediction label) of the currently screened dataset, including agonist and inhibitor, and divide into the head and tail classes.

After the preprocessing, we obtain the specific dataset for long-tailed DTI mechanism prediction. 13,389 and 829 triplets, of which triplet format is (drug SMILES, target sequence, DTI mechanism class), are obtained for the processed GtoPdb and ChEBML. The former two are used as the model input and the latter as the ground truth label.

Table 5: Detailed information of three datasets of drug-target interaction mechanism prediction.

| Dataset | Reference | #Class | #Samples | #Total Number |
|---------|-----------|--------|----------|---------------|
| GtoPdb | (Harding et al., 2018) | Inhibitor | 5,672 | 13,381 |
| | | Agonist | 3,234 | |
| | | Antagonist | 2,829 | |
| | | Allosteric modulator | 581 | |
| | | Channel blocker | 499 | |
| | | Activator | 358 | |
| | | None | 165 | |
| | | Gating inhibitor | 43 | |
| ChEBML | (Mendez et al., 2018) | Inhibitor | 421 | 829 |
| | | Antagonist | 213 | |
| | | Agonist | 181 | |
| | | Channel blocker | 12 | |
| | | Allosteric modulator | 2 | |
| GtoPdb-GPCRs | (Harding et al., 2018) | Agonist | 2606 | 5,319 |
| | | Antagonist | 2399 | |
| | | Non-target | 314 | |

## B.2 DATASET DETAILS

In this paper, three DTI mechanism datasets are used to evaluate the efficacy of the proposed method. Appendix Table 5 provides a detailed presentation of each dataset, including the types of DTI mechanisms, the sample number with different mechanisms, and the total number of the whole dataset. All these datasets exhibit the long-tailed distribution. All DTI mechanism samples are structured into triplet scheme (drug SMILES, target sequence, DTI mechanism class).

Furthermore, the relationships among the three datasets are clarified as follows: (1) GtoPdb serves as the training set for 5-fold cross-validation and internal testing. (2) ChEMBL acts as an entirely independent external test set, evaluated using the models trained on GtoPdb, ensuring no overlap with GtoPdb and thus guaranteeing fairness in testing. (3) GtoPdb-GPCRs, a subset of GtoPdb related to the GPCRs target family, is used to validate the generalization ability of the proposed method.

## B.3 PERFORMANCE COMPARISON OF DEEPPURPOSE

DeepPurpose (Huang et al., 2020a) supports training of customized DTI prediction models by implementing different compound and protein encoders and over 50 neural architectures. Here we adopt the pair combination of 7 drug encoders and 7 target encoders to display the performance.

Appendix Table 6 presents the prediction performance of all the 49 combinations on the GtoPdb dataset, which is an extension of Table 1. All the results are presented as "mean$_{\pm \text{standard deviation}}$" and the best one shown in Table 1 is the combination of "Daylight + AAC" architecture.

## B.4 IMPLEMENTATION DETAILS

To accurately evaluate model performance and prevent overfitting, we use 5-fold cross-validation to evaluate the prediction performance. The Cross Entropy loss function is used to measure model performance in the expertise training stage. Adam optimizer is adopted to optimize all of the parameters in the model with a learning rate of 0.001. The batch size is setting to 32. The training epoch for each expertise model is 1500 at most. The drug and protein embedding size $d_{\mathcal{M}}, d_{\mathcal{T}}$ is fixed to 128. The initialized atom feature vectors are described with DGL-LifeSci (Li et al., 2021) package with the embedding size $f_{\mathcal{M}} = 74$. The embedding size $f_{\mathcal{T}}$ of the initialized protein feature vectors is setting to 1200. The number of GCN, CNN, MLP layers $L_{\mathcal{M}}, L_{\mathcal{T}}, L$ are all fixed to 3. The reward and penalty score $\beta_R, \beta_P$ are both setting to 1.

Table 6: Prediction performance of DeepPurpose on the GtoPdb dataset.

| Methods | | Metrics | |
| --- | --- | --- | --- |
| Drug Encoder | Target Encoder | Accuracy↑ | F1 score↑ |
| Morgan | AAC | $0.901_{\pm 0.005}$ | $0.775_{\pm 0.021}$ |
| | Conjoint_triad | $0.898_{\pm 0.001}$ | $0.787_{\pm 0.026}$ |
| | PseudoAAC | $0.814_{\pm 0.005}$ | $0.656_{\pm 0.041}$ |
| | Quasi-seq | $0.800_{\pm 0.004}$ | $0.632_{\pm 0.024}$ |
| | CNN | $0.882_{\pm 0.005}$ | $0.738_{\pm 0.041}$ |
| | CNN_RNN | $0.881_{\pm 0.003}$ | $0.735_{\pm 0.012}$ |
| | Transformer | $0.869_{\pm 0.006}$ | $0.721_{\pm 0.025}$ |
| Pubchem | AAC | $0.904_{\pm 0.003}$ | $0.785_{\pm 0.019}$ |
| | Conjoint_triad | $0.906_{\pm 0.007}$ | $0.790_{\pm 0.029}$ |
| | PseudoAAC | $0.837_{\pm 0.007}$ | $0.707_{\pm 0.029}$ |
| | Quasi-seq | $0.809_{\pm 0.005}$ | $0.643_{\pm 0.028}$ |
| | CNN | $0.898_{\pm 0.004}$ | $0.764_{\pm 0.022}$ |
| | CNN_RNN | $0.895_{\pm 0.006}$ | $0.770_{\pm 0.017}$ |
| | Transformer | $0.880_{\pm 0.005}$ | $0.746_{\pm 0.022}$ |
| Daylight | AAC | $0.907_{\pm 0.008}$ | $0.804_{\pm 0.031}$ |
| | Conjoint_triad | $0.903_{\pm 0.009}$ | $0.788_{\pm 0.031}$ |
| | PseudoAAC | $0.827_{\pm 0.009}$ | $0.702_{\pm 0.032}$ |
| | Quasi-seq | $0.801_{\pm 0.003}$ | $0.659_{\pm 0.025}$ |
| | CNN | $0.899_{\pm 0.005}$ | $0.776_{\pm 0.023}$ |
| | CNN_RNN | $0.896_{\pm 0.004}$ | $0.778_{\pm 0.021}$ |
| | Transformer | $0.879_{\pm 0.004}$ | $0.757_{\pm 0.017}$ |
| rdkit | AAC | $0.902_{\pm 0.006}$ | $0.789_{\pm 0.029}$ |
| | Conjoint_triad | $0.904_{\pm 0.007}$ | $0.796_{\pm 0.021}$ |
| | PseudoAAC | $0.831_{\pm 0.006}$ | $0.690_{\pm 0.015}$ |
| | Quasi-seq | $0.797_{\pm 0.010}$ | $0.654_{\pm 0.041}$ |
| | CNN | $0.896_{\pm 0.010}$ | $0.762_{\pm 0.048}$ |
| | CNN_RNN | $0.898_{\pm 0.003}$ | $0.789_{\pm 0.021}$ |
| | Transformer | $0.884_{\pm 0.006}$ | $0.759_{\pm 0.012}$ |
| CNN | AAC | $0.893_{\pm 0.010}$ | $0.748_{\pm 0.035}$ |
| | Conjoint_triad | $0.895_{\pm 0.006}$ | $0.770_{\pm 0.024}$ |
| | PseudoAAC | $0.820_{\pm 0.006}$ | $0.641_{\pm 0.016}$ |
| | Quasi-seq | $0.789_{\pm 0.010}$ | $0.584_{\pm 0.019}$ |
| | CNN | $0.899_{\pm 0.003}$ | $0.768_{\pm 0.037}$ |
| | CNN_RNN | $0.869_{\pm 0.015}$ | $0.719_{\pm 0.030}$ |
| | Transformer | $0.884_{\pm 0.004}$ | $0.751_{\pm 0.009}$ |
| CNN_RNN | AAC | $0.893_{\pm 0.008}$ | $0.762_{\pm 0.029}$ |
| | Conjoint_triad | $0.890_{\pm 0.011}$ | $0.768_{\pm 0.045}$ |
| | PseudoAAC | $0.813_{\pm 0.010}$ | $0.668_{\pm 0.042}$ |
| | Quasi-seq | $0.795_{\pm 0.006}$ | $0.627_{\pm 0.030}$ |
| | CNN | $0.886_{\pm 0.006}$ | $0.771_{\pm 0.015}$ |
| | CNN_RNN | $0.878_{\pm 0.003}$ | $0.749_{\pm 0.015}$ |
| | Transformer | $0.874_{\pm 0.002}$ | $0.743_{\pm 0.012}$ |
| Transformer | AAC | $0.901_{\pm 0.008}$ | $0.789_{\pm 0.013}$ |
| | Conjoint_triad | $0.901_{\pm 0.013}$ | $0.795_{\pm 0.029}$ |
| | PseudoAAC | $0.817_{\pm 0.004}$ | $0.681_{\pm 0.028}$ |
| | Quasi-seq | $0.818_{\pm 0.010}$ | $0.668_{\pm 0.047}$ |
| | CNN | $0.891_{\pm 0.002}$ | $0.759_{\pm 0.026}$ |
| | CNN_RNN | $0.907_{\pm 0.005}$ | $0.792_{\pm 0.011}$ |
| | Transformer | $0.884_{\pm 0.007}$ | $0.751_{\pm 0.032}$ |

All experiments are conducted by PyTorch on a single NVIDIA A6000 Tensor Core GPU (48GB) and Intel(R) Xeon CPU with 24 cores and 500G memory. The whole training time for all 28 sub-tasks of GtoPdb dataset is about 8 hours, and the inference time for test set is about 2 minutes.

## B.5    METRIC DETAILS

This task involves a long-tailed multi-classification problem, where the ratio between the most frequent (head) class and the least frequent (tail) class is 132:1 (Figure 3). In such highly imbalanced scenarios, it is crucial to use evaluation metrics that provide a holistic view of model performance rather than favoring dominant classes. Given the total number of classes $N$, the accuracy and F1 score are explained in detail, highlighting why the F1 score is more appropriate for long-tailed classification tasks.

**Accuracy.** Accuracy is one of the most common metrics for classification problems and is defined as the ratio of correctly classified instances to the total number of instances.

$$\text{Accuracy} = \frac{\sum_{n=1}^{N}(TP_n + TN_n)}{\sum_{n=1}^{N}(TP_n + TN_n + FP_n + FN_n)}.$$

Here, $TP$ (True Positives) and $TN$ (True Negatives) are the correctly classified positive and negative samples for class $n$, respectively, while $FP$ (False Positives) and $FN$ (False Negatives) are the misclassified instances for class $n$. Although accuracy is straightforward, it suffers in imbalanced datasets. In long-tailed distributions, accuracy is dominated by the head classes because the model tends to classify most instances as the majority class, thus overestimating performance while ignoring the minority classes.

**F1 score.** The F1 score (Sokolova & Lapalme, 2009) is the harmonic mean of precision and recall, effectively balancing these two metrics. In multi-class settings, precision and recall are calculated for each class, and the corresponding F1 score is then obtained. Finally, the F1 scores for all classes are averaged, which can be formulated as:

$$\text{Precision}_n = \frac{TP_n}{TP_n + FP_n}, \quad \text{Recall}_n = \frac{TP_n}{TP_n + FN_n},$$

$$\text{F1 score}_n = 2 \times \frac{\text{Precision}_n \times \text{Recall}_n}{\text{Precision}_n + \text{Recall}_n}, \quad \text{F1 score} = \frac{1}{N}\sum_{n=1}^{N}\text{F1 score}_n.$$

The F1 score ranges from 0 to 1, where a higher value indicates a better balance between precision and recall. Unlike accuracy, the F1 score is less sensitive to class imbalance, making it particularly suitable for long-tailed tasks, as it ensures both head and tail classes contribute to the final evaluation.

**Importance of F1 score in Long-Tailed Tasks.** In long-tailed distributions, head classes often dominate accuracy due to their large sample size. However, for real-world problems like DTI mechanism prediction, correct predictions for tail classes are often more valuable. The F1 score provides a more nuanced view by equally weighing the importance of each class through the balance of precision and recall. This makes the F1 score the most critical metric in evaluating model performance under imbalanced conditions.

## C    DETAIL OF DISCUSSION

Detailed discussion about the proposed method is provided as follows. First, an additional discussion on specific experimental results is presented, focusing on the method's superiority and applicability. Then, the motivation behind TED-DTI is introduced from a life sciences perspective, which arises from the synergistic relationship between the advancements in neuroscience and artificial intelligence. Next, the advantage of promoting collaborative drug discovery is discussed in detail. Finally, the common solution for precious DTI methods are presented as the supplement of main paper.

### C.1    ADDITIONAL EXPERIMENTAL DISCUSSION

**Why TED-DTI excellent?** In general, TED-DTI addresses the challenges of long-tailed DTI mechanism prediction from three perspectives: (1) The divide-and-conquer strategy is adopted to decompose the original task into sub-tasks, ensuring that head classes do not dominate the resources of tail

classes and reducing the difficulty of the original task; (2) The introduce of class $\mathcal{N}_\otimes$ determines the decision boundaries of the sub-task for class $\mathcal{A}$ and $\mathcal{B}$, and generate more robust representation for tail classes by supplementing new samples from class $\mathcal{N}_\otimes$; (3) Experimental results in Table 1, Figure 4a and Figure 4b demonstrate not only the best performance but also enormous optimization potential. Additionally, Figure 4c exhibits the capability of TED-DTI to generalize to other tasks of different scales or domains.

**Solution for performance bottleneck.** Figure 4b shows the performance relationship between each expertise model and overall prediction, demonstrating that the prediction performance of the overall multi-classification task depends on how well each expertise model handles its assigned task. In other words, the performance of TED-DTI is limited to the expertise models. Therefore, the leading solution for the performance bottleneck is to optimize the expertise models. Optimizing the performance of machine learning tasks can start with model architecture, hyper-parameter tuning, pretraining parameter initiation, and other settings. For the graph domain, the model selection set is GNN architecture and its variants (Wu et al., 2020). In the future, the model architecture more suitable for the task can be completed automatically through strategies such as grid search (LaValle et al., 2004) in machine learning. Similarly, the hyper-parameter of models can also be fine-tuned with the same strategy.

**Potential application to other domains.** Figure 4c shows the generalization of TED-DTI on a similar long-tailed DTI task, demonstrating that applying the proposed strategy to multi-classification or multi-label problems of other domains (Krizhevsky et al., 2012; Long et al., 2015; Szegedy et al., 2013) is a potential and general solution to alleviate the existing long-tailed problems (Kang et al., 2020). Specifically, each expertise model only needs to select two classes or labels and identify their similarities and differences. Then the prediction results of all expertise models are summarized to output the final prediction result. In the design process of the expertise models, no matter which field the input data comes from (audio, image, text, molecule SMILES, or others), it can be solved by using a simple backbone of the specific field. For example, ResNet architecture (He et al., 2016) can be considered as the expertise model to solve the classification sub-task between the cat and dog in the image field. Similarly, we can use Transformer architecture (Vaswani et al., 2017) as the encoder in the text field. Undoubtedly, more complex encoder architectures can also be used, depending on the user. Therefore, the proposed strategy can be easily applied to similar problems in different fields.

**Computation Complexity.** The time complexity of our method is approximately $\mathcal{O}(N^2)$, where $N$ represents the number of classes. Consequently, the computational complexity scales quadratically with the number of mechanism classes, posing potential scalability challenges. To address this issue, we present a detailed analysis of the computational complexity, categorized into two cases based on the number of DTI mechanisms:

- For tasks with a limited number of classes (e.g., less than 20): In practical scenarios like computational biology (e.g., DTI mechanism prediction), the number of classes is inherently limited, as they represent real biological relationships. For empirical justification, in the GtoPdb dataset with 8 classes, the training time for sub-tasks is approximately 8 hours, and the inference time for the test set is about 2 minutes (lines 808-809). Each sub-task model requires only 2GB of GPU memory and can be trained in parallel. This training cost is acceptable comparable to the resource usage of multi-class baseline models. Furthermore, in comparison to the increasing computational demands of LLMs, our approach is lightweight and highly scalable. Therefore, our method is well-suited to most real-world tasks with limited class numbers.

- For tasks with a large number of classes: In cases where the number of classes exceeds practical thresholds, we propose two strategies to control computational complexity: (a) Using the method as an auxiliary to multi-class classification models: Instead of solving all sub-tasks, our approach can serve as an auxiliary component to refine predictions on ambiguous or long-tailed classes. This significantly reduces the number of required sub-task models while maintaining performance. (b) Constructing models only for "neighboring" classes: By leveraging class correlations, we can limit sub-task construction to semantically or structurally related classes, reducing both memory and time requirements.

## C.2 Neuroscience-inspired Motivation

The idea of tri-comparison expertise decision strategy origins from the evolution of human olfactory system. Over millions of years, the human olfactory system has gradually evolved to have the ability to perceive and distinguish various smells. Specifically, each olfactory receptor in the olfactory epithelium can recognize specific chemical structural features in odor molecules, that is, an odor molecule is decomposed into different chemical structural features and binds to specific receptors respectively, and then multiple electrical signals produced by the olfactory sensory neurons are transmitted to the high-level cognitive area for centralized decision-making and finally produce a judgment on the smell (Malnic et al., 1999). As the number and diversity of olfactory receptor genes gradually increase during the evolution of human body (Niimura & Nei, 2003), the cognition of chemical structural characteristics of odor molecules is more accurate, and ultimately a more complex olfactory experience is formed. Therefore, the key to the essential function and evolvability of the olfactory system lies in the "function divide-central decision" olfactory receptor codes for odors.

Motivated by the perception process of human olfactory system, we try to mimic the "function divide-central decision" olfactory receptor codes for odors. Specifically, we disassemble the original complex multi-classification task into simple sub-tasks and assign the tasks to different expertise models, and then the class knowledge obtained by each model are ensembled together to make a comprehensive prediction for the original task. In order to comprehensively guide the own sub-task, the discrimination between the assigned task and all other tasks need to be determined, ensuring that the specific expertise can effectively achieve the assigned objective. This "function divide-central decision" mechanism not only achieves knowledge integration for multiple classes but also reduces the complexity of model learning and mitigates the effect caused by few samples.

## C.3 Promotion for Collaborative Drug Discovery

The available volume of training data in drug discovery mainly determines the quality of intelligent models (Pejo et al., 2022). Consequently, the industry is making the first steps towards federated machine learning approaches that leverage more data than a single partner (e.g., a pharmaceutical company) (Hanser, 2023).

The proposed tri-comparison expertise decision strategy can apply the mode of federated learning, which can effectively protect the security of data and models. On the one hand, each expertise model is only stored in the local terminal of each partner. The central processor only collects the prediction results of the expertise models for sub-tasks, and there is no exchange of specific information such as model parameters and gradients in this process; on the other hand, each expertise model only needs a moderate amount of labeled data from two different classes or labels during training, and there is no need for interaction of training data between expertise models. Even if the attackers get access to the interface of the central processor, they cannot obtain any specific information about data and models. Consequently, the strategy can effectively enhance the security of the DTI application systems and protect sensitive information.

## C.4 Common Solution for Previous DTI Methods

In general, the common solution of previous DTI methods to address the DTI prediction problem is to adopt two encoders to translate the chemical information of drugs and proteins into feature representation and then output the interaction type with a decoder network after processing the combined feature.

In this process, the main difference is the encoder architecture for processing drug and target information. The biochemical structure of drugs can be represented by 1D SMILES (Weininger, 1988) and 2D molecule graphs. Therefore, various 1D fingerprint generators and 2D encoder architectures are used to extract the feature information of drugs, such as Morgan fingerprint (Morgan, 1965), Deep Neural Network (Liu et al., 2017), Graph Neural Network (Wu et al., 2020) or Transformer (Vaswani et al., 2017). Meanwhile, since targets are usually represented by 1D protein sequences (too few data with 3D structures), it is generally encoded by architectures such as 1D Conventional Neural Network (Kiranyaz et al., 2021) or Transformer.

