# OpenReview forum: "Tri-Comparison Expertise Decision for Drug-Target Interaction Mechanism Prediction"
_ICLR.cc/2025/Conference — ICLR 2025 Conference Withdrawn Submission_

### Official Review · Reviewer_C1Wv · 2024-10-31

**Soundness:** 4
**Presentation:** 3
**Contribution:** 2
**Rating:** 5
**Confidence:** 4

**Summary:**

The paper introduces TED-DTI, a novel framework for predicting drug-target interaction (DTI) mechanisms that addresses the challenge of long-tailed class distribution in real-world drug discovery applications. The authors propose an innovative divide-and-conquer approach by decomposing the multi-class prediction task into pairwise sub-tasks, each handled by independent expertise models. A key contribution is the introduction of a tri-comparison expertise training strategy that adds a "Neither" class option to enhance discrimination between mechanism classes. The framework leverages graph neural networks for drug encoding and CNN for protein sequence encoding, combined with a class-balanced decision voting module to integrate predictions from multiple expertise models.

    The method demonstrates performance improvements on the GtoPdb and ChEMBL datasets compared to state-of-the-art methods. The approach shows special promise in handling tail classes, as evidenced by superior AUROC scores for rare mechanism classes like Gating Inhibitor.

    While the technical innovation and empirical results are compelling, there are some important limitations to consider. The computational complexity increases quadratically with the number of classes, potentially limiting scalability for larger mechanism sets. Additionally, while the "Neither" class addition is clever from a machine learning perspective, its biological significance and practical implications could be better justified.

**Strengths:**

-Novel tri-comparison strategy that effectively addresses long-tailed distribution challenges
-Strong theoretical foundation with clear connections to  neuroscience-inspired design
-Superior handling of tail classes with demonstrated improvements in rare mechanism prediction
-Potential for generalization to other interaction domains

**Weaknesses:**

- Marginal performance improvement (<1% on accuracy for both datasets)
- Limited evaluation metrics - lack of AUROC, AUPRC, and MCC metrics which are crucial for imbalanced datasets
- No comparison with 2024 state-of-the-art methods
- The "Neither" class addition lacks biological significance and may not be truly innovative
- OvO method comparisons use overly simple backbones, making the comparative experiments less meaningful
- No comparison with popular multimodal large biological models (e.g., OpenBioMed, BioMedGPT, xTrimo) in DTI prediction
- Insufficient description of important parameters and their tuning process
- Computational complexity scales quadratically with mechanism classes (O(N²)), raising scalability concerns

**Questions:**

- Why is the balanced penalty weight vector H defined for each mechanism class rather than for each sub-task?
- Is the dataset used in the study newly constructed? Is it publicly available (Will it be released)?
- For DTI baseline comparisons, were the same task decomposition and model ensemble strategies applied as in TED-DTI? This would ensure a fair comparison.
- Has the model been tested in virtual screening scenarios?
- What strategies could be employed to address the quadratic computational complexity?
- How sensitive is the model to different "Neither" class sampling strategies?

---

> ### Author Response · Authors · 2024-11-23
> **Response to Reviewer C1Wv (Part I)**
>
> Thanks for the time spent reviewing our paper, and the recognition of the novelty, significance, applicability of our work. We have carefully considered your constructive comments. Below are our point-to-point responses to your comments:
>
> > **W1:** Marginal performance improvement (<1% on accuracy for both datasets).
>
> **A:** Sorry for the confusion. The datasets used in our study are highly imbalanced, with class distribution ratios reaching up to 132:1. In such cases, accuracy primarily reflects the model's performance on the dominant (head) classes, while offering limited insight into the balanced classification across all classes. Our proposed method addresses this challenge by focusing on improving the balanced classification performance across all categories, including those in the tail. Through the introduction of the "Neither" class and other strategies, our method demonstrates superior generalization capability, as evidenced by a 13% improvement in F1 score, which is more representative of the overall performance in this highly imbalanced context.
>
> > **W2:** Limited evaluation metrics - lack of AUROC, AUPRC, and MCC metrics which are crucial for imbalanced datasets.
>
> **A:** Thanks for your concerns. Generally, our target task is a multi-class problem with a long-tailed data distribution. Therefore, we used the F1 score as it is a comprehensive metric for evaluating multi-class performance [1]. In contrast, AUROC, AUPRC, and MCC are primarily designed for evaluating binary classification tasks with imbalanced datasets [2]. Hence, the F1 score is more appropriate for capturing the nuances of this multi-class task.
>
> > **W3:** No comparison with 2024 state-of-the-art methods.
>
> **A:** We appreciate your insightful suggestions. We have now incorporated the latest methods of 2024 for comparison, and the results are as follows:
>
> | Method     | Parameter Number      | Accuracy (GtoPdb) | F1 score (GtoPdb) | Accuracy (ChEBML) | F1 score (ChEBML) |
> | ---------- | --------------------- | ----------------- | ----------------- | ----------------- | ----------------- |
> | BINDTI [3] | -  | 0.908(0.002)      | 0.806(0.028)      | 0.934(0.006)      | 0.676(0.029)      |
> | BioT5+ [4] | 252M | 0.920(0.003)      | 0.829(0.022)      | 0.954(0.002)      | 0.767(0.018)      |
> | TED-DTI    | $C_8^2\times$377K=10M | **0.924(0.004)**  | **0.834(0.012)**  | **0.961(0.003)**  | **0.789(0.040)**  |
>
> As shown in the table above, our method still demonstrates significant advantages over the supervised-based BINDTI and cross-modal pre-trained LLM BioT5+. Notably, despite having a parameter size significantly smaller than BioT5+ (25 times fewer parameters), our method still outperforms BioT5+ in terms of performance. In the revised version, we have provide these additional experimental results in Table 1 & 2.
>
> > **W4:** The "Neither" class addition lacks biological significance and may not be truly innovative.
>
> **A:** Thanks for useful suggestions. Our method indeed holds biological significance. It addresses ambiguous samples that traditional methods struggle with and enhances the model’s ability to handle complex biological data. For example, we analyze the relationship between agonists and activators:
>
> - Agonists directly bind and activate receptors, while activators enhance biological responses by amplifying the action of other molecules.
> - The traditional OvO strategy, with its strict "either-or" classification, often misclassifies ambiguous samples, limiting the model’s learning. The "Neither" class improves this by better handling such samples, avoiding oversimplified errors, and allowing for more accurate biological distinctions. This differentiation aids in understanding drug mechanisms and improves the model's fit for complex biological systems.
>
> In the future, we will provide more details on biological significance.
> > **W5:** OvO method comparisons use overly simple backbones, making the comparative experiments less meaningful.
>
> **A:** Sorry for the misunderstanding. The introduction of OvO methods is to validate the improvements brought by the innovation of our method over the OvO strategy. To ensure fairness, our proposed TED-DTI uses the exact same model architecture as the GCN-based OvO in Table 1, with the only difference being that the prediction classes for each sub-task changed from 2 to 3 (due to the introduction of class Neither). Therefore, we did not use overly simple backbones only for OvO baselines. We have clarified this point in the revised version (line 431).
> > **W6:** No comparison with popular multimodal large biological models (e.g., OpenBioMed, BioMedGPT, xTrimo) in DTI prediction.
>
> **A:** Thanks for suggestions. In **W3**, we have compared our method with BioT5+ [4], a cross-modal text-based large model recently accepted at ACL 2024. Despite having a parameter size significantly smaller than BioT5+ (25 times fewer parameters), our method still outperforms BioT5+ in terms of performance.

---

> ### Author Response · Authors · 2024-11-23
> **Response to Reviewer C1Wv (Part II)**
>
> > **W7:** Insufficient description of important parameters and their tuning process.
>
> **A:** We apologize for the confusion. All details of important parameters have been provided (Appendix Section B.4). We will move more important descriptions into the main body for clear reading.
> > **W8:** Computational complexity scales quadratically with mechanism classes (O(N²)), raising scalability concerns.
>
> **A:** Thanks for deeper consideration. To address this concern, we provide a detailed analysis of the computational complexity of our method, divided into two cases based on the number of DTI mechanisms:
>
> **1) For tasks with a limited number of classes (e.g., less than 20):**
>
> The time complexity of our method is $\mathcal{O}(N^2)$, where $N$ is the number of classes. In practical scenarios like computational biology (e.g., DTI mechanism prediction), the number of classes is inherently limited, as they represent real biological relationships.
>
> For empirical justification, the training time for sub-tasks is $\sim$8 hours (lines 972-975). Each sub-task model requires only 2GB of GPU memory and can be trained in parallel. This training cost is acceptable comparable to the resource usage of these baselines. Therefore, our method is well-suited to most real-world tasks with limited class numbers.
>
> **2) For tasks with a large number of classes:**
>
> In cases where the number of classes exceeds practical thresholds, we propose two strategies to control computational complexity: **(a) Auxiliary to multi-class classification models:** Instead of solving all sub-tasks, our approach can serve as an auxiliary component to refine predictions on ambiguous or long-tailed classes. This significantly reduces the number of required sub-task models while maintaining performance. **(b) Constructing only for "neighboring" classes:** By leveraging class correlations, we can limit sub-task construction to semantically or structurally related classes, reducing both memory and time requirements.
>
> In the revised version, we have provide a more detailed discussion (lines 1056-1079).
> > **Q1:** Why is the balanced penalty weight vector H defined for each mechanism class rather than for each sub-task?
>
> **A:** Thanks for valuable comment. The balanced penalty weight vector $\mathbf{H}$ is only used during the inference phase and is independent of the training process. Since this is a multi-class classification problem with class imbalance, the balance coefficient is applied at the final class level to maximize predictive gains.
> > **Q2:** Is the dataset used in the study newly constructed? Is it publicly available (Will it be released)?
>
> **A:** Yes, the datasets in this work were extracted from public datasets, which will be released in the final version.
> > **Q3:** For DTI baseline comparisons, were the same task decomposition and model ensemble strategies applied as in TED-DTI? This would ensure a fair comparison.
>
> **A:** Yes, we applied the same strategies for the OvO-based methods in the baselines, whereas the other baselines did not require task decomposition.
> > **Q4:** Has the model been tested in virtual screening scenarios?
>
> **A:** Thanks for consideration regarding applicability. We have collaborated with a biotechnology company and validated TED-DTI's screening accuracy and efficiency in real-world scenarios. However, we have not yet identified drug candidates that have successfully passed wet lab experiments.
> > **Q5:** What strategies could be employed to address the quadratic computational complexity?
>
> **A:** Thanks for concern. The computational complexity can be controlled through only using the method as an auxiliary to multi-class classification models, or constructing models only for "neighboring" classes (please see **W8**).
> > **Q6:** How sensitive is the model to different "Neither" class sampling strategies?
>
> **A:** Thanks for comment. For a comprehensive comparison, we adopt three sampling strategies for class "Neither": Cluster-based Sampling, Active Learning-based Sampling, and Random Sampling (Appendix Section A.1). Among these strategies, Random Sampling performed the worst (0.789$\pm$0.040), as the selected "Neither" samples lacked representativeness. In contrast, Active Learning-based Sampling (0.812$\pm$0.017) showed a 5.5% improvement over Random Sampling, as it actively chooses the most uncertain samples for the "Neither" class. In the final version, we will provide a detailed discussion of these different sampling strategies.
>
> **References**
>
> [1] Sokolova M, Lapalme G. A systematic analysis of performance measures... . Information processing & management, 2009.
>
> [2] Chicco, D., Jurman, G. The advantages of the Matthews correlation coefficient... . *BMC Genomics*, 2020.
>
> [3] Peng, Lihong, et al. BINDTI: A bi-directional Intention network... . *IEEE Journal of Biomedical and Health Informatics, 2024.*
>
> [4] Pei, Qizhi, et al. BioT5+: Towards Generalized Biological Understanding... . *ACL 2024 (Findings).*

---

> ### Author Response · Authors · 2024-11-28
> **Kind reminder to expect your feedback**
>
> Dear Reviewer C1Wv,
>
> I hope this message finds you well. I would like to kindly follow up regarding our revised manuscript and the response to your valuable comments. We have made significant updates based on your constructive suggestions, and we would greatly appreciate it if you could review our responses at your earliest convenience.
>
> Thank you again for your time and consideration. We look forward to your feedback.
>
> Best regards,\
> Submission6294 Authors

---

> > ### Author Response · Authors · 2024-12-02
> > **Gentle Reminder for Reviewer C1Wv: Review Period Closing Soon**
> >
> > Dear Reviewer **C1Wv**,
> >
> > We kindly remind you that the review period will conclude in **less than 24 hours**, with December 2nd being the last day for reviewers to post messages to the authors.
> >
> > In our previous responses, we have thoroughly addressed all of your concerns and questions. We sincerely hope you can provide feedback on our responses, as your recognition is crucial to us.
> >
> > Once again, we deeply appreciate your time, effort, and thoughtful review of our work.
> >
> > Best regards,\
> > Submission6294 Authors

---

### Official Review · Reviewer_xMku · 2024-11-01

**Soundness:** 2
**Presentation:** 3
**Contribution:** 1
**Rating:** 3
**Confidence:** 5

**Summary:**

This paper improves the one-vs-one method on long-tailed multi-class classification problem by expanding the sub-tasks from binary classification to tri-comparison, with an additional ``neither" class. The method is evaluated on both internal and external dataset, and shown better performance. Noticeable improvement is seen especially on extremely tail class.

**Strengths:**

What this paper focuses is an important but long-been-overlooked problem. In my opinion, performance on tail classes can easily be overwhelmed in common metrics, such as micro-averaging ones. I am glad to see this paper makes some analysis on extremely tail class, thus highlighting the problem.

The experimental design is solid, especially including external dataset evaluation and generalization validation. The dataset size is relatively small, but I understand it is limited by available public data.

The paper is well organized and presented. It is generally easy to understand, but you should try to make things more concise, to avoid put important information in supplementary.

**Weaknesses:**

I think adding an additional "neither" class is somewhat rough. You are using more data compared with binary classification, but it is still imbalanced in each sub-tasks. Besides, classes have complex relationships between them, so simply putting a highly-correlated class into "neither" may not seem a good idea. Also from a practical perspective, both the proposed method and one-vs-one generates too many sub-tasks. It is expensive to train all these models, and makes it infeasible to scale to larger number of classes.

Also, the method seems not highly coupled with DTI. Actually the only things related to DTI is the two encoders, but the architecture is widely used. So instead of limiting to DTI, authors should try to apply their method to more domains. This also solves the problem of limited dataset size.

**Questions:**

The paper trains one GCN(for Drug), one CNN(for Target), and one MLP for each sub-task, which incurs a significant additional computational overhead. Have the authors considered using a shared Drug Encoder and a shared Target Encoder for each sub-task? Can you discuss the tradeoffs between using separate vs shared encoders, including any potential impacts on performance or training time?

How the authors obtained the class-balanced weight vector H.

In line 302, how the authors performed the pre-processing. Specifically, how do you handle the missing data? Are they simply been filtering out? Is there any else filtering criteria?

In the experimental section, only the results for the Gating inhibitor are presented. However, data for Allosteric modulators, Channel blockers, and Activators are also limited and should be analyzed as well.

In Table 2, the addition of class “neither” leads to a notable increase in the F1 score only in ChEMBL, rising from 0.648 to 0.789. Can you explain the reason? Also, the accuracy only improves from 0.955 to 0.961. Why is the difference between these two metrics? Can you discuss the implications of these differences for the model's performance on different datasets or class distributions?

---

> ### Author Response · Authors · 2024-11-23
> **Response to Reviewer xMku (Part I)**
>
> Thanks for the time spent reviewing our paper, and the recognition of the novelty, significance, presentation ability, applicability of our work. We have carefully considered your constructive comments. Below are our point-to-point responses to your comments:
> > **W1:** The raising problem of introducing class Neither: (a) You are using more data compared with binary classification, but it is still imbalanced in each sub-tasks. (b) Classes have complex relationships between them, so simply putting a highly-correlated class into "neither" may not seem a good idea. (c) It is expensive to train all these models, and makes it infeasible to scale to larger number of classes.
>
> **A:** Thanks for your valuable suggestions regarding the addition of the "Neither" class.
>
> **(a)** Due to data limitations, the imbalance issue cannot be completely resolved but can be alleviated. The introduction of the "Neither" class helps reduce interference from head classes on tail classes, thereby improving the model’s precision for minority classes.
>
> **(b)** Each sub-task or expertise model is responsible for handling the relationship between the two target classes and the "Neither" class. The complex relationships between highly-correlated classes are automatically reflected in the voting process based on the collective preference of all experts, such as the total vote count for similar classes.
>
> **(c)** Using multiple sub-tasks does increase the computational cost, especially when there are a large number of categories, which can become burdensome (lines 462-472). To address this concern, we provide a detailed analysis of the computational complexity of our method, divided into two cases based on the number of DTI mechanisms: 1) For tasks with a limited number of classes (e.g., less than 20): The time complexity of our method is approximately $\mathcal{O}(N^2)$, where $N$ is the number of classes. In practical scenarios like computational biology (e.g., DTI mechanism prediction), the number of classes is inherently limited, as they represent real biological relationships. For empirical justification, the training cost is acceptable comparable to the resource usage of multi-class baseline models (lines 808-809). Therefore, our method is well-suited to most real-world tasks with limited class numbers. 2) For tasks with a large number of classes, the complexity can be controlled through:
> * Using the method as an auxiliary to multi-class classification models. Instead of solving all sub-tasks, our approach can serve as an auxiliary component to refine predictions on ambiguous or long-tailed classes. This significantly reduces the number of required sub-task models while maintaining performance.
> * Constructing models only for "neighboring" classes. By leveraging class correlations, we can limit sub-task construction to semantically or structurally related classes, reducing both memory and time requirements.
>
> In the final version, we will provide a more detail supplement.
> > **W2:** The proposed method seems not highly coupled with DTI. Authors should try to apply their method to more domains.
>
> **A:** Thanks for your concern. Firstly, the problem we aim to address is not the traditional binary classification of DTI, but a deeper exploration of the drug-target response mechanisms, which is a multi-class problem with a long-tailed distribution. The method we proposed is specifically designed to solve this long-tailed problem, which has significant practical implications. In addition, we have discussed the potential applications to other domains (e.g., computer vision and natural language processing) in Appendix Section C.1 (lines 894-907). The application strategy is quite straightforward, demonstrating a general solution to alleviate existing long-tailed problems. In the future, we plan to apply the proposed strategy to more domains.
> > **Q1:** (a) Consider shared encoders for each sub-task? (b) Discuss the tradeoffs between separate vs shared encoders (impacts on performance or training time)?
>
> **A:** Thanks for your insightful suggestions. **(a)** Each trained model for a sub-task represents expertise in that task, specifically designed to achieve precise tri-classification for the two assigned categories under any circumstances. The model parameters are unique and irreplaceable for that specific task, which is why, in principle, they cannot be shared. **(b)** Using separate encoders allows each model to specialize, ensuring high precision, particularly for long-tailed or non-linear class boundaries. The tradeoff is higher computational cost and training time. Instead, shared encoders reduce computational burden by leveraging shared features but may struggle with distinct class boundaries, potentially lowering performance. Hence, separate encoders provide task-specific precision and robustness, essential for imbalanced, non-overlapping class distributions. In the final version, we will provide additional discussion for this point.

---

> ### Author Response · Authors · 2024-11-23
> **Response to Reviewer xMku (Part II)**
>
> > **Q2:** How the authors obtained the class-balanced weight vector $\mathbf{H}$?
>
> **A:** Thanks for your comment. The class-balanced weight vector $\mathbf{H}$ weights the importance of each class to ensure that the contribution of all classes is fairly evaluated in the system, similar to the distribution of probabilities and weights across different states in a thermodynamic system.
>
> Specifically, $\mathbf{H}$ is calculated by $w_c=\frac{\frac{1}{N_c}}{\sum_{k=1}^C \frac{1}{N_k}}$, where $w_c$ represents the weight for class $c$ ; $N_c$ represents the number of samples in class $c$ ; $C$ represents the total number of classes. In the final version, we will supplement this definition.
>
> > **Q3:** (a) In line 302, how the authors performed the pre-processing. (b) Specifically, how do you handle the missing data? Are they simply been filtering out? (c) Is there any else filtering criteria?
>
> **A:** Thank you for your concern. **(a)** Detailed data preprocessing procedures are provided in Appendix Section B.1 (lines 740-755). We used the RDKit package to verify the validity of drug SMILES and obtained protein sequences via the SwissProt target protein identifier. **(b)** Invalid or missing data was excluded during preprocessing. **(c)** No, there is no other filtering criteria due to the data format.
>
> > **Q4:** In the experimental section, data for Allosteric modulators, Channel blockers, and Activators are also limited and should be analyzed as well.
>
> **A:** Thank you for pointing this out. Our intention was to highlight this minority class, which accounts for only 0.3% of the dataset, to evaluate our model's generalizability under extreme conditions. The results clearly demonstrate that the model is both effective and robust. Additionally, we have also validated the model on other minority classes, achieving improvements in ROC-AUC scores of 0.9%-4.1% compared to the second-best method. We will include additional results in the final version.
>
> > **Q5:** (a) In Table 2, the addition of class "neither" leads to a notable increase in the F1 score only in ChEMBL, rising from 0.648 to 0.789. Can you explain the reason? (b) Also, the accuracy only improves from 0.955 to 0.961. Why is the difference between these two metrics? (c) Can you discuss the implications of these differences for the model's performance on different datasets or class distributions?
>
> **A:** Thanks for your insightful question. First, we would like to clarify that our goal is to solve a long-tailed multi-classification task.
>
> **(a)** GtoPdb represents an internal test set, which is more susceptible to overfitting, resulting in a modest improvement of only 2.21%. In contrast, ChEMBL serves as a more challenging external test set, where we observe a significant improvement of 12.88%. This demonstrates the superior generalization ability of our method.
>
> **(b)**  For an extremely imbalanced multi-classification task (132:1), accuracy is simply the ratio of correct predictions to total predictions, which can be dominated by the majority class in imbalanced datasets. In contrast, the F1 score reflects the balance between precision and recall, providing a more comprehensive measure of performance, especially for minority classes.
>
> **(c)** Here, we provide an overall summary of the impact of datasets and metrics on performance:
>
> * **Scope of Evaluation.** To clarify our experimental setup: during the 5-fold cross-validation training, we only used the GtoPdb training set. For testing, we evaluated the metrics on both the GtoPdb test set (internal test) and the entire ChEMBL dataset (external test) using the trained models. As a result, the modest improvement observed on the GtoPdb test set (F1 score increased by 2.21%) could be partially attributed to overfitting, whereas the performance improvement on ChEMBL (F1 score increased by 12.88%) demonstrates the strong generalization ability of our method.
> * **Metric Significance.** This task involves a long-tailed multi-classification problem, where the ratio between the most frequent (head) class and the least frequent (tail) class is 132:1 (Figure 3). Therefore, in such a highly imbalanced scenario, accuracy primarily reflects the performance on the head classes and does not provide a balanced view of model effectiveness. In contrast, the F1 score balances precision and recall across all classes, making it the most critical evaluation metric. As shown in Table 1, while the accuracy shows only slight improvements, the significant enhancement in F1 score better represents the overall significance of our findings.

---

> > ### Comment · Reviewer_xMku · 2024-12-03
> >
> > I thank the authors for their response. For the weaknesses, I cannot be persuaded. The authors also admit that the imbalance problem is not resolved but only alleviated by their method. For me, it may not be attractive enough since it brings heavy overhead and adds the complexity for the training and deployment.
> >
> > With respect to W2, I would like to emphasize that, the method seems not using any features that are specific to the DTI problem (except the encoder architecture), so it could have been developed as a general method and applied to many areas. If so, the contribution would be stronger. I am sorry I have to keep the score as it is.

---

> > > ### Author Response · Authors · 2024-12-04
> > > **Response to New Comment from Reviewer xMku**
> > >
> > > Thanks for your feedback. Here are our responses to your remaining concerns:
> > >
> > > > **Q1:** For the weaknesses, I cannot be persuaded. The authors also admit that the imbalance problem is not resolved but only alleviated by their method.
> > >
> > > **A:** In fact, the mentioned imbalance problem (i.e., long-tailed task) is a highly significant research area in artificial intelligence [1]. To date, no work has claimed to have fully resolved this challenge. Instead, research in this field is a continuous progression, and expecting a complete resolution is neither realistic nor should it be considered a weakness that undermines the contributions we have made in this domain.
> > >
> > > **References**\
> > > [1] Zhang, Yifan, et al. Deep long-tailed learning: A survey. IEEE Transactions on Pattern Analysis and Machine Intelligence (TPAMI), 2023.
> > >
> > > > **Q2:**  For me, it may not be attractive enough since it brings heavy overhead and adds the complexity for the training and deployment.
> > >
> > > **A:** We have provided detailed explanations regarding time and GPU usage, along with reasonable strategies to reduce complexity. Compared to LLMs with parameter counts 25 times greater than our method, we believe the claim of "heavy overhead" is unwarranted. If possible, please specify any long-tailed biological scenarios where our method would be impractical or exceed existing resource limitations.
> > >
> > > > **Q3:** With respect to W2, I would like to emphasize that, the method seems not using any features that are specific to the DTI problem (except the encoder architecture), so it could have been developed as a general method and applied to many areas. If so, the contribution would be stronger.
> > >
> > > **A:** The DTI mechanism task aims to determine the action type of a drug on a target protein, which naturally exhibits a long-tail distribution as a multi-class problem. Therefore, proposing a method to address this long-tail issue is highly significant—a point you previously acknowledged with the comment, "an important but long-been-overlooked problem." We find it perplexing why our contribution in this context is now perceived as minimal.
> > >
> > > Regarding the model architecture, utilizing drug and protein features typically occurs within the encoders, a practice consistently adopted in previous methods (e.g., Nature Machine Intelligence, ACL Findings, Bioinformatics).
> > >
> > > Finally, both in our manuscript and in our responses to your comments, we have provided analyses and discussed the potential applications of this method as a general framework that can be applied across various domains.

---

> ### Author Response · Authors · 2024-11-28
> **Kind reminder to expect your feedback**
>
> Dear Reviewer xMku,
>
> I hope this message finds you well. I would like to kindly follow up regarding our revised manuscript and the response to your valuable comments. We have made significant updates based on your constructive suggestions, and we would greatly appreciate it if you could review our responses at your earliest convenience.
>
> Thank you again for your time and consideration. We look forward to your feedback.
>
> Best regards,\
> Submission6294 Authors

---

> > ### Author Response · Authors · 2024-12-02
> > **Gentle Reminder for Reviewer xMku: Review Period Closing Soon**
> >
> > Dear Reviewer **xMku**,
> >
> > We kindly remind you that the review period will conclude in **less than 24 hours**, with December 2nd being the last day for reviewers to post messages to the authors.
> >
> > In our previous responses, we have thoroughly addressed all of your concerns and questions. We sincerely hope you can provide feedback on our responses, as your recognition is crucial to us.
> >
> > Once again, we deeply appreciate your time, effort, and thoughtful review of our work.
> >
> > Best regards,\
> > Submission6294 Authors

---

### Official Review · Reviewer_Ev5A · 2024-11-02

**Soundness:** 2
**Presentation:** 2
**Contribution:** 3
**Rating:** 5
**Confidence:** 4

**Summary:**

This paper addresses the multi-class problem of Drug-Target Interaction (DTI) under long-tail distribution and proposes an algorithm based on the divide-and-conquer strategy—Tri-Comparison Expertise Decision (TED-DTI). Compared to other DTI methods and long-tailed learning-based methods, TED-DTI achieves better accuracy and F1 score on the DTI task, particularly with the AUROC metric for long-tail categories being higher than that of other algorithms. The TED-DTI algorithm is an innovative upgrade based on the One-vs-One algorithm, with the core idea of extending the classification task from A/B to A/B/Neither. The authors demonstrate through ablation experiments that this approach effectively improves the model's performance on the test set.

**Strengths:**

The TED-DTI algorithm is an innovative upgrade based on the One-vs-One algorithm, with the core idea of extending the classification task from A/B to A/B/Neither. The authors demonstrate through ablation experiments that this approach effectively improves the model's performance on the test set.And the authors explain that the introduction of 'Neither' may enhance model performance for the following reasons:(1) The design of class N⊗ improves the discrimination between mechanism classes and provides more expressive representations. (2)Supplementing a large number of samples from the 'Neither' class aids in feature learning.  The multi-class problem under long-tail distribution is a common issue, not only in the DTI context. It is expected that applying this algorithm to other scenarios should also lead to improved model performance, particularly in making more accurate predictions for minority classes. However, this paper focuses solely on the DTI problem and does not provide experimental applications in other scenarios, which is a regrettable omission.

**Weaknesses:**

I believe that the TED-DTI algorithm can enhance the model's performance on long-tail categories in the DTI multi-class problem. However, the innovation of TED is not significant; it appears to be a minor modification of the One-vs-One approach, essentially changing the original binary classification problem into a three-class problem. Additionally, the introduction of 'Neither' is similar to the 'Rest' in One-vs-Rest, suggesting that TED seems to be a combination of One-vs-One and One-vs-Rest. From this perspective, the authors should briefly introduce One-vs-Rest in the related work section and discuss the connections and differences among the three approaches in subsequent sections. Furthermore, One-vs-Rest should be included in the method comparison (Table-1).

**Questions:**

The paper has some imprecise parts; here are a few:

1. As stated in line 244, the loss function formula for each tri-comparison expertise model shows that the loss is the mean of the cross-entropy for N categories. However, each tri-comparison expertise model is a three-class model and is trained separately. Therefore, its loss should be the mean of the cross-entropy for the three categories, not N. Although both 3 and N are constants and do not actually affect the model training, correcting this would make the paper more precise. I recommend that the authors to clarify if N should be replaced with 3 in the formula, or if there is some other reason for using N that is not explained in the current text.

2. Table 1 (lines 342 and 343) presents a comparison between two OvO methods and TED-DTI. However, the backbone models of these two OvO methods are SVM-based and GCN-based, respectively, which are not strictly consistent with the backbone model of the TED-DTI method. Therefore, the conclusion stated in lines 413-415, "Compared with OvO methods which also adopt the divide-and-conquer strategy, TED-DTI significantly exceeds all the OvO baselines," cannot be drawn from this comparison. However, this conclusion can be supported by the ablation experiments described in Table 2.  So I suggest that the authors either use consistent backbone models across all compared OvO methods, or explicitly acknowledge and discuss the impact of different backbones on the performance comparison.

3. The authors selected 829 samples from the ChEMBL dataset as an independent test set. However, it should be analyzed whether there is any overlap between these samples and the training set (GtoPdb). If  overlap exists, the overlapping samples should be removed. The authors should explicitly state in the paper whether they checked for and removed any overlapping samples, and if so, describe the process they used.

Additionally, for the experiments, the following should be addressed:

1. The GtoPdb-GPCRs dataset has a total of 5,111 samples, while the GtoPdb dataset has a total of 13,381 samples. Is the GtoPdb-GPCRs (5,111 samples) included within the 13,381 samples of GtoPdb, or is it additional? Is is suggested that the authors clarify in the paper the relationship between the GtoPdb and GtoPdb-GPCRs datasets, specifically whether GtoPdb-GPCRs is a subset of GtoPdb or a separate dataset.
2. As mentioned in line 354, the authors performed 5-fold cross-validation on the training set. So, i'm confused and would appreciate it if the authors could explicitly describe in the paper how the metrics for the ChEBML dataset in Table 1 were calculated. Are the metrics for various algorithms on the ChEBML dataset the mean values of the metrics from the five models obtained through 5-fold cross-validation on GtoPdb, or were they derived in another way?

Finally, providing open-source code and data would be beneficial.

---

> ### Author Response · Authors · 2024-11-23
> **Response to Reviewer Ev5A (Part I)**
>
> Thanks for the time spent reviewing our paper, and the recognition of the novelty, applicability of our work. We have carefully considered your constructive comments. Below are our point-to-point responses to your comments:
>
> > **W:** (a) The connections and differences among the One-vs-One approach, One-vs-Rest approach, and the proposed Tri-comparison. (b) Additional experimental comparison with One-vs-Rest.
>
> **A:** We appreciate your insightful suggestions.
>
> **(a)** TED-DTI addresses key challenges faced by OvR and OvO, such as misclassification of unrelated samples and exacerbated data imbalance. Its theoretical advantages in generalization and long-tailed distribution handling make it especially effective for multi-class tasks like drug-target interaction prediction, setting it apart with superior robustness and adaptability. Specifically:
>
> | Method  | Similarity                                                   | Difference                                                   |
> | ------- | ------------------------------------------------------------ | ------------------------------------------------------------ |
> | OvR     | (1) Decomposes the multi-class task into multiple binary classification sub-tasks. (2) Utilizes existing binary classification models for multi-class prediction. | **(1) Exacerbates data imbalance:** OvR compares one class (positive) against all others (negative), significantly increasing the imbalance, especially under long-tailed distributions. **(2) Strict class boundaries:** Positive and negative samples are strictly divided, leading to rigid boundaries prone to overfitting, especially on head classes, limiting generalization for tail classes. **(3) No explicit handling of unrelated samples:** Unable to address noise or unrelated data effectively, potentially impacting performance. |
> | OvO     | (1) Similar to OvR, limits each sub-task to involve only two classes, reducing the complexity of each model. (2) Can utilize smaller training sets for faster training and inference. | **(1) Ignores unrelated samples:** OvO only establishes decision boundaries between two classes, lacking explicit modeling for unrelated or noisy samples, potentially leading to misclassification of these samples. **(2) Limited decision boundaries:** Each classifier operates independently, making it harder to benefit from global relationships among classes, thus limiting overall performance. |
> | TED-DTI | (1) Similar to OvO, each classification involves limited classes, reducing the complexity of the multi-class problem. (2) Captures decision boundaries for head classes effectively. | **(1) "Neither" class:** TED-DTI explicitly introduces a "Neither" class, modeling unrelated samples and improving robustness to noise and long-tailed distributions, avoiding common misclassification issues in OvR and OvO. **(2) Improved decision boundaries:** The "Neither" class expands decision spaces between categories, mitigating data imbalance and rigid boundary issues, and theoretically optimizing the generalization error bound, particularly under long-tailed distributions. |
>
> In the revised version, we have expanded the discussion on the One-vs-Rest (OvR) approach in the "Related Work" section and provided a more comprehensive analysis of the similarities and differences among the three methods (lines 119-143).
>
> **(b)** To quantitatively compare the performance of the three methods, we provide the results of OvR (using the same sub-task model structure as OvO and TED-DTI) as follows. These results demonstrate that our method improves the F1 score (ChEMBL) by 39% and 22% compared to OvR and OvO, respectively. In the revised version, we have added the results in Table 1.
>
> | Method  | Accuracy (GtoPdb) | F1 score (GtoPdb) | Accuracy (ChEBML) | F1 score (ChEBML) |
> | ------- | ----------------- | ----------------- | ----------------- | ----------------- |
> | OvR     | 0.887 (0.010)     | 0.732 (0.049)     | 0.910 (0.015)     | 0.566 (0.051)     |
> | OvO     | 0.916 (0.004)     | 0.812 (0.030)     | 0.955 (0.007)     | 0.648 (0.129)     |
> | TED-DTI | 0.924 (0.004)     | 0.834 (0.012)     | 0.961 (0.003)     | 0.789 (0.040)     |

---

> ### Author Response · Authors · 2024-11-23
> **Response to Reviewer Ev5A (Part II)**
>
> > **Q1:** Loss function formula (line 244) for each tri-comparison expertise model is imprecise.
>
> **A:** Thank you for your thorough review. Indeed, we mistakenly presented the formula in line 244; the value of $N$ should be replaced with 3, as each expert model corresponds to three classes. The correct formula for each sub-task should be $\mathcal{L}=-\frac{1}{3} \sum_{n=1}^3 p_n \log \left(\hat{p}_n\right)$. In the revised version, we have revised this mistake (lines 254-256).
>
> > **Q2:** The implementation details of OvO methods for performance comparison with TED-DTI in Table 1.
>
> **A:** We are sorry for the confusion. First of all, the reason for introducing OvO-based methods is that the TED-DTI method shares certain similarities with OvO, and we made innovations based on this. Specifically, the details and analysis of the OvO methods are as follows:
>
> - The GCN-based OvO is a "degeneration" of TED-DTI. In the GCN-based OvO, each sub-model's task is to classify two DTI mechanism classes, excluding the new "Neither" class that we introduced. To ensure fairness, the architecture of the subtask model in GCN-based OvO is identical to that in TED-DTI, except for the difference in the final prediction layer output.
> - As for the SVM-based OvO, it is introduced to highlight the advancement of the subtask model and its contribution to the overall performance improvement.
> - In Table 2, since the GCN-based OvO represents the result of the ablation experiment that removes the "Neither" class, we have directly used the GCN OvO results from Table 1.
>
> In the revised version, we have supplemented the description for clarification (line 431).
>
> > **Q3:** The details of ChEMBL dataset: (a) Has ChEMBL dataset  been checked for and removed any overlapping samples of GtoPdbs; (b) Describe the process they used.
>
> **A:** Thanks for your comment. **(a)** Yes, ChEMBL serves as a completely independent and external test set, ensuring no overlap with the GtoPdb dataset, which guarantees fairness in testing. **(b)** As a fully independent and external test set, ChEMBL only use in the test stage to demonstrate the generalizability and robustness of the proposed method ($\sim$13% improvements with a quite small variance). Specifically, for the 5-fold cross-validation training, we only use the GtoPdb training set; for the testing, we report the metrics on both GtoPdb test set (internal test) and the full ChEMBL dataset (external test) through these trained models. In the revised version, we have clarified this point (lines 892-897).
>
> > **Q4:** The relationship between the GtoPdb and GtoPdb-GPCRs datasets, specifically whether GtoPdb-GPCRs is a subset of GtoPdb or a separate dataset.
>
> **A:** Thanks for pointing this out. In fact, GtoPdb-GPCRs is a subset of GtoPdb (lines 317-318). The purpose of creating GtoPdb-GPCRs is to validate the generalizability of the TED-DTI method on similar problems of different scales (Fig. 4c & lines 425-431). In addition, GPCRs represents an important family of human target proteins, and this dataset can serve as a well-organized resource for future research. In the revised version, we have clarified this point (line 355 & lines 892-897).
>
> > **Q5:** Are the metrics for the ChEBML dataset in Table 1 the mean values of the metrics from the five models obtained through 5-fold cross-validation on GtoPdb, or were they derived in another way?
>
> **A:** Yes,  the reported metrics for ChEBML (Table 1) are obtained through the 5-fold cross-validation models on GtoPdb.
>
> > **Q6:** Providing open-source code and data would be beneficial.
>
> **A:** Thanks for your comment. We have provided part of the code and dataset in the Supplementary Materials, and the complete code and dataset will be made available in the final version.

---

> ### Author Response · Authors · 2024-11-28
> **Kind reminder to expect your feedback**
>
> Dear Reviewer Ev5A,
>
> I hope this message finds you well. I would like to kindly follow up regarding our revised manuscript and the response to your valuable comments. We have made significant updates based on your constructive suggestions, and we would greatly appreciate it if you could review our responses at your earliest convenience.
>
> Thank you again for your time and consideration. We look forward to your feedback.
>
> Best regards,\
> Submission6294 Authors

---

> > ### Author Response · Authors · 2024-12-02
> > **Gentle Reminder for Reviewer Ev5A: Review Period Closing Soon**
> >
> > Dear Reviewer **Ev5A**,
> >
> > We kindly remind you that the review period will conclude in **less than 24 hours**, with December 2nd being the last day for reviewers to post messages to the authors.
> >
> > In our previous responses, we have thoroughly addressed all of your concerns and questions. We sincerely hope you can provide feedback on our responses, as your recognition is crucial to us.
> >
> > Once again, we deeply appreciate your time, effort, and thoughtful review of our work.
> >
> > Best regards,\
> > Submission6294 Authors

---

### Official Review · Reviewer_ZQ3u · 2024-11-03

**Soundness:** 3
**Presentation:** 2
**Contribution:** 2
**Rating:** 6
**Confidence:** 2

**Summary:**

This paper introduces TED-DTI, a novel framework for drug-target interaction (DTI) mechanism prediction that addresses the challenge of long-tailed class distributions. The key innovation is a tri-comparison expertise decision approach that (1) decomposes the multi-class problem into pairwise sub-tasks using divide-and-conquer, (2) introduces a third "Neither" class for enhanced discrimination between similar mechanism classes, and (3) employs a class-balanced decision voting module for final predictions. The method is extensively evaluated on three datasets and demonstrates significant improvements over state-of-the-art methods, particularly for tail classes.

**Strengths:**

- A novel tri-comparison expertise training strategy
- A class-balanced decision voting module that effectively combines expertise predictions with weighted rewards/penalties
- Comprehensive empirical validation and thorough ablation studies demonstrating the importance of key components

**Weaknesses:**

- The paper would benefit from stronger theoretical justification for why the tri-comparison approach works better than binary classification.
- How generalizable it is when dealing with less or more classes?
- Additional evaluation/analysis on empirical and theoretical computational costs. Specifically, since each sub-task requires a separate model, potentially making the overall system resource-intensive.

**Questions:**

See above.

---

> ### Author Response · Authors · 2024-11-23
> **Response to Reviewer ZQ3u (Part I)**
>
> Thanks for the time spent reviewing our paper, and the recognition of the novelty, significance of our work. We have carefully considered your constructive comments. Below are our point-to-point responses to your comments:
>
> > **W1:** Theoretical analysis for why the tri-comparison approach works better than binary classification.
>
> **A:** Thanks for highlighted suggestion. The tri-comparison strategy presents a holistic and robust solution for multi-class classification, particularly in long-tailed tasks such as DTI mechanism prediction. By integrating principles from decision boundary theory and error decomposition, this approach progressively enhances classification performance and generalization ability.
>
> In traditional binary classification for multi-class problems, decision boundaries (e.g., $f_{i,j}(x)$ for classes $C_i$ and $C_j$) often encounter noise and bias caused by overlapping regions from unrelated samples. This issue is especially prominent in real-world tasks with long-tailed distributions. The tri-comparison strategy addresses this challenge by introducing a "Neither" class, with a new decision boundary $f_{\text{Neither}}(x)$. This additional boundary explicitly identifies unrelated samples, creating a three-region space partition: $ \mathbb{R}^d = \\{ x : f_{i}(x) > f_{\text{Neither}}(x) \\} \cup \\{ x : f_{j}(x) > f_{\text{Neither}}(x) \\} \cup \\{ x : f_{\text{Neither}}(x) > \max(f_{i}(x), f_{j}(x)) \\} $. This refinement in decision boundaries reduces the noise caused by ambiguous samples, ensuring clearer separation between classes and laying a foundation for improved classification accuracy.
>
> Building on this enhanced boundary framework, the tri-comparison approach further reduces classification errors through a more nuanced error decomposition. In binary classification, the overall error $\epsilon_{\text{binary}}$ is dominated by the false negative rate of minority classes and the false positive rate of majority classes. By explicitly isolating unrelated samples into the "Neither" class, the classification error is redefined as $\epsilon_{\text{tri}} = \epsilon_{\text{false positive}} + \epsilon_{\text{false negative}} + \epsilon_{\text{Neither}}$. This separation reduces the overlap between positive and negative classes, significantly lowering $\epsilon_{\text{false positive}}$ and $\epsilon_{\text{false negative}}$, and consequently decreasing the total error. The tri-comparison strategy thus moves beyond simple noise reduction, actively addressing imbalances in class representation to improve classification reliability.
>
> In the final version, these theoretical analysis will be supplemented to support the observed empirical improvements in tasks such as DTI mechanism prediction.
>
> > **W2:** The generalizability to deal with less or more classes.
>
> **A:** Thank you for your valuable comment. Our proposed TED-DTI method demonstrates a strong generalizability to alleviate existing long-tailed problems. Specifically:
>
> * **Potential DTI mechanism tasks of different scales.** Generally, the overall framework of the proposed stragegy, that is, "Task Decomposition - Tri-Comparison Expertise Training - Overall Decision Voting", is clear and does not rely on a specific network architecture. Therefore, our strategy can be quickly adapted to solve tasks of varying scales. In addition, due to the representation of real biological relationships, the number of classes in computational biology tasks (including DTI mechanisms) is typically limited to fewer than 20, avoiding the risk of complexity explosion.
> * **Extension to broader domains.** Due to the powerful and straightforward tri-comparison strategy, TED-DTI demonstrates strong potential for extension to broader research domains, such as computer vision and natural language processing (Appendix Section C.1, lines 894-907).
>
> * **Experimental validation.** We have discussed its generalizability on similar tasks with less classes (lines 425-431). As shown in Fig. 4c, our proposed method achieves significant improvements.
>
> In the final version, we will include a more detailed discussion on generalizability.

---

> ### Author Response · Authors · 2024-11-23
> **Response to Reviewer ZQ3u (Part II)**
>
> > **W3:** Additional evaluation/analysis on empirical and theoretical computational costs.
>
> **A:** Thank you for deeper consideration. To address this concern, we provide a detailed analysis of the computational complexity of our method, divided into two cases based on the number of DTI mechanisms:
>
> **1) For tasks with a limited number of classes (e.g., less than 20):**
>
> The time complexity of our method is approximately $\mathcal{O}(N^2)$, where $N$ is the number of classes. In practical scenarios like computational biology (e.g., DTI mechanism prediction), the number of classes is inherently limited, as they represent real biological relationships.
>
> For empirical justification, in the GtoPdb dataset with 8 classes, the training time for sub-tasks is approximately 8 hours, and the inference time for the test set is about 2 minutes (lines 808-809). Each sub-task model requires only 2GB of GPU memory and can be trained in parallel. This training cost is acceptable comparable to the resource usage of multi-class baseline models. Furthermore, in comparison to the increasing computational demands of LLMs, our approach is lightweight and highly scalable. Therefore, our method is well-suited to most real-world tasks with limited class numbers.
>
> **2) For tasks with a large number of classes:**
>
> In cases where the number of classes exceeds practical thresholds, we propose two strategies to control computational complexity:
>
> - **(a) Using the method as an auxiliary to multi-class classification models:**
>   Instead of solving all sub-tasks, our approach can serve as an auxiliary component to refine predictions on ambiguous or long-tailed classes. This significantly reduces the number of required sub-task models while maintaining performance.
> - **(b) Constructing models only for "neighboring" classes:**
>   By leveraging class correlations, we can limit sub-task construction to semantically or structurally related classes, reducing both memory and time requirements.
>
> In the final version, we will provide a more detailed discussion on computational complexity.

---

> > ### Comment · Reviewer_ZQ3u · 2024-11-23
> >
> > Thank you for your reply. Most of my concerns have been solved and I would like to keep my overall positive score.

---

> > > ### Author Response · Authors · 2024-11-28
> > >
> > > Thanks for your positive recognition. If there’s anything further we can address to fully meet your expectations, please let us know—we’re happy to improve further.

---

### Official Review · Reviewer_QcRP · 2024-11-04

**Soundness:** 3
**Presentation:** 2
**Contribution:** 2
**Rating:** 6
**Confidence:** 4

**Summary:**

The authors address the problem of predicting DTI mechanisms by developing a decision method called TED-DTI using deep learning in the following way:
1. for every pair of mechanisms, training a "two-vs-rest" classifier for the mechanisms plus an "other" class made up of examples from the rest of the mechanisms, and
1. at inference time, ensembling the predictions by a novel class-balanced voting mechanism.

**Strengths:**

- novel two-vs-rest pairwise classifier and class-balanced penalization for voting
- performance improvements over many baselines in Table 1
- demonstrated improvements over both one-vs-one classification and standard voting in Table 2

**Weaknesses:**

- Makes comparisons to older (ca. 2020) DTI prediction models, there are many newer ones in the
- Except for the F1 score on ChEMBL, the gains of TED-DTI over the next best model are very modest.
- Table 1 inaccurately reports the lift $\Delta$, e.g. the ChEMBL F1 score should be $12.88\\%$ since $0.789/0.699 = 1.1288$
- one-vs-one (a.k.a. all-vs-all) classification is a long-standing technique in multi-class classification, and is covered in classic ML texts like Bishop's Pattern Recognition and Machine Learning. Authors should have cited this and other earlier papers, e.g. [1] and [2]. I'm also surprised by the claim that two-vs-rest classification hasn't been reported in the literature before, but after a bit of searching I also couldn't find a reference.

[1] Allwein, Erin L., Robert E. Schapire, and Yoram Singer. "Reducing multiclass to binary: A unifying approach for margin classifiers." Journal of machine learning research 1.Dec (2000): 113-141.
[2] Wu, Ting-Fan, Chih-Jen Lin, and Ruby Weng. "Probability estimates for multi-class classification by pairwise coupling." Advances in Neural Information Processing Systems 16 (2003).

**Questions:**

- How did you choose the baselines models used for comparison? There are many newer DTI prediction models that could have been used.
- Why wasn't the data in Fig 4a and 4c presented as a table like Table 1, along with stdevs and lift? It looks like TED-DTI might have achieved roughly 1.5% improvement over LADE and DrugBAN, which is again modest.

---

> ### Author Response · Authors · 2024-11-23
> **Response to Reviewer QcRP (Part I)**
>
> Thanks for the time spent reviewing our paper, and the recognition of the novelty, significance of our work. We have carefully considered your constructive comments. Below are our point-to-point responses to your comments:
>
> > **W1:** New DTI prediction models for experimental comparison.
>
> **A:** We appreciate your insightful suggestions. We have now incorporated the latest 2024 methods for comparison, and the results are as follows:
>
> | Method      | Parameter Number          | Accuracy (GtoPdb) | F1 score (GtoPdb) | Accuracy (ChEBML) | F1 score (ChEBML) |
> | ----------- | ------------------------- | ----------------- | ----------------- | ----------------- | ----------------- |
> | BINDTI [1]  | -                         | 0.908 (0.002)     | 0.806 (0.028)     | 0.934 (0.006)     | 0.676 (0.029)     |
> | BioT5+  [2] | 252M                      | 0.920 (0.003)     | 0.829 (0.022)     | 0.954 (0.002)     | 0.767 (0.018)     |
> | TED-DTI     | $C_8^2 \times $377K = 10M | **0.924 (0.004)** | **0.834 (0.012)** | **0.961 (0.003)** | **0.789 (0.040)** |
>
> As shown in the table above, our method still demonstrates significant advantages over the supervised-based BINDTI and cross-modal pre-trained LLM BioT5+. Notably, despite having a parameter size significantly smaller than BioT5+ (25 times fewer parameters), our method still outperforms BioT5+ in terms of performance. In the final version, we will provide these additional experimental results.
>
> > **W2:** Except for the F1 score on ChEMBL, the other improvements are very modest.
>
> **A:** We are sorry for the confusion. The improvements observed in Table 1 are not as modest as they may appear at first glance. We provide a detailed explanation from two perspectives: the scope of evaluation and the significance of metrics.
>
> * **Scope of Evaluation.** To clarify our experimental setup: during the 5-fold cross-validation training, we only used the GtoPdb training set. For testing, we evaluated the metrics on both the GtoPdb test set (internal test) and the entire ChEMBL dataset (external test) using the trained models. As a result, the modest improvement observed on the GtoPdb test set (F1 score increased by 2.21%) could be partially attributed to overfitting, whereas the performance improvement on ChEMBL (F1 score increased by 12.88%) demonstrates the strong generalization ability of our method. This external testing challenge has been noted and acknowledged by Reviewer *xMku*.
> * **Metric Significance.** This task involves a long-tailed multi-classification problem, where the ratio between the most frequent (head) class and the least frequent (tail) class is 132:1 (Figure 3). Therefore, in such a highly imbalanced scenario, accuracy primarily reflects the performance on the head classes and does not provide a balanced view of model effectiveness. In contrast, the F1 score balances precision and recall across all classes, making it the most critical evaluation metric. As shown in Table 1, while the accuracy shows only slight improvements, the significant enhancement in F1 score better represents the overall significance of our findings.
>
> In addtion, our current approach combines classical machine learning techniques with simple deep networks, utilizing fixed hyperparameters across all expertise models (sub-tasks). Fine-tuning the hyperparameters for each sub-task in the future is expected to yield even greater enhancements (see Appendix Section C.1). In the final version, we will include additional discussions on this topic.
>
> > **W3:** The improvement of ChEMBL F1 score is inaccurate.
>
> **A:**  We apologize for this error and thank you for your careful attention. The F1 score improvement on the ChEMBL dataset should be 12.88% (although this does not affect the conclusion of a significant improvement). In the final version, we will correct this point.
>
> > **W4:**  Citation for basic solutions. (a) Additional citations for one-vs-one classification. (b) Two-vs-rest classification hasn't been reported in the literature before.
>
> **A:** Thank you for your suggestion. **(a)** We will add additional references to this classic and long-standing algorithm in the main text. **(b)** In fact, when we first came up with this idea, we also considered whether there were related works, but we were unable to find any. We believe this approach is both interesting and effective, as combining improved classical machine learning strategies with simple deep networks can surpass the existing complex network architectures, including LLMs. In the final version, we will supplement the relevant descriptions.

---

> ### Author Response · Authors · 2024-11-23
> **Response to Reviewer QcRP (Part II)**
>
> > **Q1:** How to choose the baselines models used for comparison? There are many newer DTI prediction models that could have been used.
>
> **A:** Thank you for your comment. To clarify, this paper proposes a Tri-comparison method based on innovations in the OvO approach to address the challenge of predicting DTI mechanisms with long-tailed distributions. We selected these baseline models for comparison based on the following three reasons:
>
> - DTI: Building on the deep exploration of DTI tasks, we are the first to propose a task for predicting the proper mechanisms of drugs act with the targets. Since there are no existing methods for this task, we compare against advanced DTI baselines;
> - LTL: As our task is inherently a long-tailed multi-class problem, we include widely used baselines specifically designed to address long-tailed issues;
> - OvO: Since our method is an innovation and extension of the OvO strategy, we compare it with its original versions.
>
> Additionally, we provide results from recent baselines, as shown in W1. In the final version, we will enhance this discussion to further support our choices.
>
> > **Q2:**  Why not Fig 4a and 4c presented as the tables (with stdevs and lift)? The improvements of Fig 4a is again modest.
>
> **A:** Thanks for valuable suggestions. Fig. 4a shows the ROC scores for evaluating on the "Gating Inhibitor" class. Our proposed TED-DTI method consistently achieves ROC-AUC score above 0.90 (i.e., 0.914±0.013), while the other baselines exhibit large variances, indicating that previous methods are highly sensitive to changes in data distribution/fold and show poor generalizability. The specific experimental results of  Fig. 4a and 4c are provided as follows, which will be supplemented in the final version:
>
> * The ROC-AUC results of Fig. 4a for few-sample class:
>
> | Methods          | ROC-AUC     |
> | ---------------- | ----------- |
> | DeepPurpose      | 0.865±0.155 |
> | DeepConv-DTI     | 0.893±0.093 |
> | MolTrans         | 0.843±0.067 |
> | DrugBAN          | 0.899±0.043 |
> | CB               | 0.875±0.083 |
> | Focal Loss       | 0.823±0.115 |
> | LADE             | 0.900±0.085 |
> | ESQL             | 0.849±0.095 |
> | Balanced Softmax | 0.766±0.076 |
> | Weighted Softmax | 0.861±0.100 |
> | LDAM             | 0.875±0.107 |
> | GCL              | 0.857±0.082 |
> | SVM-based OvO    | 0.755±0.170 |
> | GCN-based OvO    | 0.858±0.082 |
> | TED-DTI          | 0.914±0.013 |
> | $\Delta$         | +1.56%      |
>
> * The Accuracy and F1 score results of Fig. 4c for the generalizability to deal with GPCR task:
>
> | Methods  | Accuracy      | F1 score      |
> | -------- | ------------- | ------------- |
> | GPCR ML  | 0.820 (0.017) | 0.748 (0.024) |
> | TED-DTI  | 0.889 (0.006) | 0.877 (0.011) |
> | $\Delta$ | +8.42%        | +17.25%       |
>
> **References**
>
> [1] Peng, Lihong, et al. BINDTI: A bi-directional Intention network for drug-target interaction identification based on attention mechanisms. *IEEE Journal of Biomedical and Health Informatics, 2024.*
>
> [2] Pei, Qizhi, et al. BioT5+: Towards Generalized Biological Understanding with IUPAC Integration and Multi-task Tuning. *ACL 2024 (Findings).*

---

> > ### Comment · Reviewer_QcRP · 2024-11-26
> >
> > Thank you for your response and addressing my concerns and questions.
> > * Thank you for providing more modern DTI model comparisons, including an LLM.
> > * Your point about the F1 score on the held-out ChemBL subset being significantly better is well-taken. In order to help the reader understand what kind of a lift 12.88% is, it would be useful to see the delta in confusion matrices between the top two classes.
> > * Thank you for clarifying the choice of models used for comparison.
> > * Thank you for providing the tables associated with Fig 4, along with stdevs. It's interesting that for ROC-AUC, the mean of LADE and DrugBAN are very close to being within one stdev of the mean of TED-DTI, and of course the mean of TED-DTI is within one stdev of the means of both LADE and DrugBAN, respectively. Was this the best example across all small classes?
> >
> > I've raised my score.

---

> > > ### Author Response · Authors · 2024-11-28
> > >
> > > Thanks for your positive recognition and further suggestions. Here are our responses:
> > >
> > > > **Q1:** ... understand what kind of a lift 12.88% is, it would be useful to see the delta in confusion matrices ...
> > >
> > > **A:** Thanks for your valuable suggestion. For Accuracy and F1 score, we have provided a detailed calculation process with the elements from the confusion matrices (Appendix Section B.5). Additionally, for a better understanding of how the improvements were made, we have provided the confusion matrix comparison of LADE (the second best method) and TED-DTI for the minority class "Channel blocker" (only 12 test samples) in ChEBML dataset.
> > >
> > > First, the confusion matrix is defined as
> > > $
> > > \text{Confusion Matrix (CM)} =
> > > \begin{bmatrix}
> > > \text{TP} & \text{FP} \\\\
> > > \text{FN} & \text{TN}
> > > \end{bmatrix}.
> > > $
> > >
> > > Then, we output the confusion matrices of LADE  and our proposed TED-DTI, respectively, as bellow:
> > >
> > > $ \\text{CM}\_{\\text{ LADE}} = \begin{bmatrix}
> > > 9 & 1 \\\\
> > > 3 & 816
> > > \end{bmatrix}, \quad \\text{CM}\_\\text{ TED\-DTI} =
> > > \begin{bmatrix}
> > > 12 & 1 \\\\
> > > 0 & 816
> > > \end{bmatrix}.$
> > >
> > > Thus, for this minority class, the Accuracy of LADE/TED-DTI are **0.995/0.999**, respectively, while the F1 scores are **0.818/0.960**. This clearly demonstrates that the improvement in the F1 score arises from more accurate predictions for the minority class.
> > >
> > > At the same time, it is evident that Accuracy is primarily influenced by the correct predictions (TN) of the majority class, whereas the F1 score offers a more comprehensive evaluation of model performance in long-tail tasks.
> > >
> > > > **Q2:** It's interesting that for ROC-AUC, ... the mean of TED-DTI is within one stdev of the means of both LADE and DrugBAN, respectively.
> > >
> > > **A:** Thanks for interesting observation. The significantly large stdevs of LADE and DrugBAN indicate that these methods exhibit high sensitivity when handling extreme classes under different data distributions. This variability reflects their lack of robustness and generalization capability. Therefore, the phenomenon mentioned by the reviewer is mainly due to the significant performance fluctuations in LADE and DrugBAN, rather than evidence of stability or reliability comparable to TED-DTI.
> > >
> > > > **Q3:** Was this the best example across all small classes?
> > >
> > > **A:** Thanks for your constructive question. While the improvement (Figure 4a) may not represent the best result, it is the most crucial, as it represents the least frequent (tail) class, which accounts for only 0.3% of the entire dataset. Given that other tail classes have 4 to 14 times more samples, the improvement in the least frequent class becomes even more significant, further emphasizing TED-DTI's effectiveness in handling long-tail distribution problems. Furthermore, we have also evaluated several other tail classes, where the ROC-AUC score improvements ranged from 0.9% to 4.1% compared to the second-best method.

---

### Author Response · Authors · 2024-11-23
**Global Response**

We thank all reviewers for the time spent reviewing the paper and recognizing the advantages of our work as follows:

* **Significance**:  "an important but long-been-overlooked problem" & "Noticeable improvement" &  "The experimental design is solid" - xMku, "Strong theoretical foundation" & "Superior handling of tail classes" - C1Wv,  "performance improvements over many baselines"- QcRP,  "Comprehensive empirical validation and thorough ablation studies" - ZQ3u
* **Novelty**: "novel two-vs-rest pairwise classifier"- QcRP, "A novel tri-comparison expertise training strategy"- ZQ3u, "an innovative upgrade" - Ev5A, "Novel tri-comparison strategy" & "clever from a machine learning perspective" - C1Wv
* **Presentation Quality**: "well organized and presented" & "easy to understand" – xMku
* **Applicability**: "applying this algorithm to other scenarios" – Ev5A, "Potential for generalization to other interaction domains" – C1Wv.

We have endeavored to consider the feedback as comprehensively as possible, leading to a revision process that significantly honed the paper. We have addressed every point in our responses and are happy to follow up on any aspect during the discussion phase. Specifically, we have tackled stated weaknesses (**W**), questions (**Q**) with detailed answers (**A**).

For common reviewer concerns, we provide the following important clarifications and additions:

- **Problem addressed**: Our method aims to improve multi-class classification in highly imbalanced datasets, particularly focusing on the challenges in drug-target mechanism prediction.
- **Experimental setup**: During the 5-fold cross-validation training, we only used the GtoPdb training set. For testing, we evaluated the metrics on both the GtoPdb test set (internal test) and the entire ChEMBL dataset (external test) using the trained models.
- **Metric explanation**: Accuracy reflects overall performance but is biased toward head classes. F1 score balances precision and recall, making it the most relevant metric for our imbalanced task.
- **Additional experimental baselines**: We have included the latest 2024 baselines, including a cross-modal LLM with 252M parameters.

Finally, we would like to express our appreciation once again for the reviewers' constructive comments and careful reading, which undoubtedly lead to enhancing the quality of our work.

---

### Author Response · Authors · 2024-11-25
**Kind Request To Reviewers: We are looking forward to receiving your feedback**

Dear Reviewers,

I hope this message finds you well. First and foremost, we would like to sincerely thank you for your time and thoughtful feedback on our work. We deeply appreciate your insights and have made revisions and additions based on your suggestions.

We have provided the detailed response to each comment of all reviewers. At your convenience, we would greatly appreciate it if you could review our response and let us know if any further adjustments are needed.

We understand that the review process takes time. If there are any additional questions or if you need further clarification, please do not hesitate to reach out.

Thank you once again for your time and consideration. We look forward to hearing from you and hope for further feedback soon.

Best regards,\
Submission6294 Authors

---

### Note · Authors · 2025-01-22

I have read and agree with the venue's withdrawal policy on behalf of myself and my co-authors.